# 3D Dense Captioning beyond Nouns: A Middleware for Autonomous Driving

## Abstract

Recently, language foundation models have revolutionized many fields and how they could enable smarter and safer autonomous vehicles remains elusive. We believe one major obstacle is the lack of a comprehensive and standard middleware representation that links perception and planning. We rethink the limitations of existing middleware (e.g., 3D boxes or occupancy) and propose 3**D** d**e**nse capt**i**onin**g** beyond **n**ouns (or abbreviated as DESIGN). For each input scenario, DESIGN refers to a set of 3D bounding boxes with a language description for each. Notably, the **comprehensive** description involves not only what the box is (noun) but also its attribute (adjective), location (preposition) and moving status (adverb). We design a scalable rule-based auto-labelling methodology to generate DESIGN ground truth, guaranteeing that the middleware is **standard**. Using this methodology, we construct a large-scale dataset nuDesign based upon nuScenes, which consists of an unprecedented number of 2300k sentences. We also present an extensive benchmarking on nuDesign, featuring a model named DESIGN-former that takes multi-modal inputs and predicts reliable DESIGN outputs. Through qualitative visualizations, we demonstrate that DEISGN, as a novel 3D scene understanding middleware, has good interpretability. We release our code, data and models, hoping this middleware could trigger better autonomous driving algorithms and systems that benefit from the power of language foundation models.

## 1 Introduction

Nowadays, autonomous driving vehicles have been deployed in various restricted (e.g., port or mine area) or open (e.g., big cities with up-to-date HD maps) scenarios. But their intelligence level is still far lower than human beings, making uninterpretable driving decisions that are incredibly dangerous or violate basic social norms. Language foundation models are a potential game changer in this regard as they demonstrate impressive human-like reasoning capabilities in challenging tasks like solving math problems (Wei et al., 2022). We ask the question: Now that language foundation models have not yet enabled smarter and safer autonomous vehicles, what are the missing pieces?

We notice that mainstream autonomous driving stacks (including the latest solutions (Thrun et al., 2006; Daudelin et al., 2018; Casas et al., 2021; Hu et al., 2023; Sadat et al., 2020; Tampuu et al., 2020; Chen et al., 2023a; Hu et al., 2023; Shao et al., 2023)) are consisted of perception and planning modules. And the outputs of conventional 3D scene understanding modules extract middleware representations like 3D object bounding boxes (Shi et al., 2020) or occupancy maps (Li et al., 2023) to bridge perception and planning. These middleware representations are not readily compatible with language foundation models thus hindering the exploitation of human-like reasoning capabilities of them. As such, this study pushes forward 3D road scene understanding by defining a novel middleware named as DESIGN.

DESIGN is designed under two primary principles: (1) The middleware has to be comprehensive. The newest middleware of open-vocabulary 3D boxes (Najibi et al., 2023) is still not comprehensive enough, as it only covers object names or say nouns. To unleash the power of language foundation models, we believe a comprehensive description beyond nouns is important. (2) The middleware has to be standard. The success of existing middle representations like 3D boxes or occupancy is credited to the fact that they are standard enough to be annotated in scale by the community.

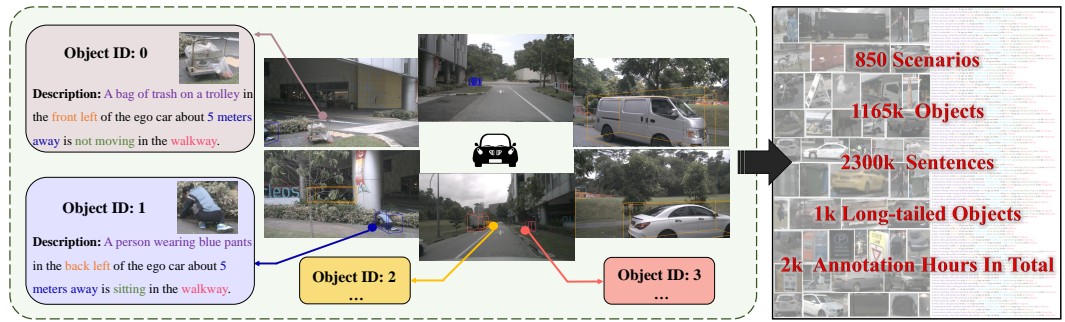

Figure 1: As existing 3D road scene understanding tasks do not provide outputs that are naturally compatible with language foundation models, we propose a new middleware representation named DESIGN. It is abbreviated for **3D** d**e**n**s**e capt**i**o**n**i**n**g beyond **n**ouns. For each 3D object box in the scene, we attach a comprehensive and standard sentence using an auto-labeling pipeline. Our nuDESIGN dataset is of an unprecedentedly large volume, as demonstrated on the right panel.

Specifically, DESIGN is short for **3D** d**e**n**s**e capt**i**o**n**i**n**g beyond **n**ouns and an intuitive demonstration of the middleware is presented in the left panel of Fig. 1. For each input driving scenario, DESIGN refers to a set of 3D bounding boxes with a natural language description for each. Instead of solely assigning an open-vocabulary object name to each 3D box, the sentence involves comprehensive information beyond nouns like attribute (adjective) or location (preposition). Going beyond nouns is critical for the goal of leveraging language foundation models. For example, for object 0, knowing the trash is *on the walkway* but not *on the lane* reduces the potential risk it may bring. Similarly, for object 1, knowing the person is *sitting* means that she does not have a good view of the surrounding so the vehicle should watch closely on her. Interestingly, it is known that language foundation models are good at making reasoning like *a person that is sitting does not have a good view*.

Apart from being comprehensive, a good middleware should be standard so that the community can annotate using a common format and down-streaming algorithms can expect a common input. So we take the perspective of procedural generation and design an automatic labelling pipeline (Fig. 3). The sentence template of DESIGN is demonstrated in Fig. 1 and different fields of the template are generated by corresponding modules. The appearance description of the object, which consists primarily adjectives and nouns, is generated by a learned captioning module that involves GPT-3.5 rewriting and human feedback. Other object states, which consists primarily adverbs and prepositions, are generated by summarizing conventional 3D scene understanding middle representation that already exists in mainstream datasets (i.e., nuScenes). We note that being standard naturally brings the advantage of scalability and we build an extremely large dataset named nuDESIGN (Fig. 2).

As demonstrated in the right panel of Fig. 1, there are as many as 2300k object descriptions in our nuDESIGN dataset, all in the comprehensive and standard description formats mentioned above. A natural question is how difficult it is to extract this new middleware representation from sensory inputs. Through extensive benchmarking, we show that existing dense captioning methods all perform poorly on this challenging task and we credit this to the fact that they fail to leverage the rich commonsense knowledge in language foundation models. To this end, we propose DESIGN-former, which is a novel query-based architecture that fine-tune adapters imposed upon the rich representation of LLaMa (Touvron et al., 2023a). Both quantitative and qualitative results demonstrate the surprising effectiveness of DESIGN-former to predict this new middleware representation.

To summarize, our contributions are: (1) As existing 3D road scene understanding tasks are not readily compatible with language foundation models, we propose a new middleware named DESIGN, short for **3D** d**e**n**s**e capt**i**o**n**i**n**g beyond **n**ouns. (2) We propose a scalable automatic labelling pipeline that involves GPT-3.5 rewriting, human feedback and rule-based summarization. The result is an unprecedentedly large dataset called nuDESIGN, containting 2300k descriptions for 1165k objects in 850 scenes. (3) We also contribute DESIGN-former, which is a novel query-based network architecture that fine-tunes adapters on LLaMa. Thanks to the power of foundation models, our proposal significantly out-performs existing dense captioning baselines. (4) We also present a pilot study that reveals the impact of this new middleware representation on down-streaming tasks.

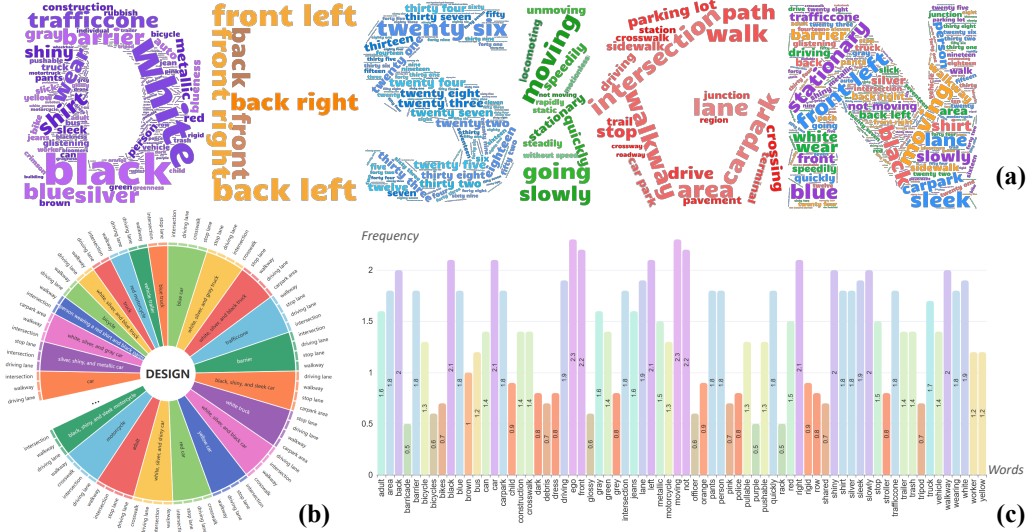

Figure 2: The properties of nuDESIGN. (a) The word cloud distribution of nuDESIGN. Each part represents the different components of sentence annotation and the last column represents the overall annotation. Note that in the third column we convert the Arabic numerals to text. (b) The distribution of Object's appearance and road map. (c) Statistics of the top-70 word frequency.

## 2  RELATED WORK

### 2.1  LINGUISTIC CAPABILITY IN AUTONOMOUS DRIVING

The success of large language models (LLM) (Ouyang et al., 2022; Taori et al., 2023; Touvron et al., 2023a;b) inspires the exploration of linguistic capability of autonomous driving vehicles. Generally, the combination of natural language and autonomous driving includes two aspects: natural language understanding (NLU) and natural language generation (NLG). NLU refers to the ability of a self-driving vehicle to interpret human language, which can be used to understand and execute language commands (Contributors, 2023; Fu et al., 2023; Park et al., 2023; Shah et al., 2023). NLG refers to the ability of a system to generate natural language responses, which can be used to provide verbal instructions to passengers (Kim et al., 2019), or generate natural language explanations about the vehicle's performance (Jin et al., 2023; Kim et al., 2018). In this work, we focus on the agent-level natural language generation in driving scenarios, aiming to enrich the feature representation of the perception module beyond traditional category-based methods to better help the subsequent decision-making system.

### 2.2  DENSE CAPTIONING

As a fundamental problem that connects computer vision and natural language processing, visual captioning tasks (Vinyals et al., 2015; Xu et al., 2015) aim to describe the objects and their relationship of a given image in natural language, which usually takes a visual encoder to extract the visual features and a language decoder to generate language sentences. Similarly, dense captioning (Johnson et al., 2016) aims at generating detailed descriptions for local regions, which usually needs to detect visual regions and generate descriptions for these regions. Most closely related to our work is Scan2Cap (Chen et al., 2021b), which introduces 3D captioning task that tries to describe each object of a 3D scene. However, such indoor literature (Chen et al., 2023b; Cai et al., 2022; Chen et al., 2021a) is difficult to be applied to driving scenarios, which contains complex dynamic outdoor objects. To bridge the gap, we introduce dense driving captioning task, which describes the rich representation of each object in driving scenes. We hope modeling the perception module in a linguistic paradigm can introduce the advantages of natural language to autonomous driving, leading to a stronger scene understanding ability.

| Dataset | Publication | Input Modality | Sentences Number | Level | Goal | Appearance | Direction | Distance | Motion | Road Map |
|---|---|---|---|---|---|---|---|---|---|---|
| BDD-X (Kim et al., 2018) | ECCV2018 | C | 26k | Scene | Captioning | | | | | |
| HAD (Kim et al., 2019) | CVPR2019 | C | 45k | Scene | Grounding | | | | ✓ | |
| Cityscapes-Ref (Vasudevan et al., 2018) | CVPR2018 | C | 30k | Object | Grounding | | | | | |
| Talk2Car (Deruyttere et al., 2019) | IJCNLP2019 | C+L | 12k | Object | Grounding | | | | | |
| Refer-KITTI (Wu et al., 2023) | CVPR2023 | C+L | 818 | Object | Grounding | ✓ | ✓ | | | |
| DRAMA (Malla et al., 2023) | WACV2023 | C | 102k | Object | Captioning | ✓ | | ✓ | ✓ | |
| nuScenes-QA (Qian et al., 2023) | arxiv2023 | C+L | 460k | Object | QA | | ✓ | | ✓ | |
| **DESIGN** | | C+L | **2300k** | **Object** | **Dense Captioning** | ✓ | ✓ | ✓ | ✓ | ✓ |

Table 1: Comparison with existing driving language datasets. The C and L in the third column represent the Camera and LiDAR, respectively

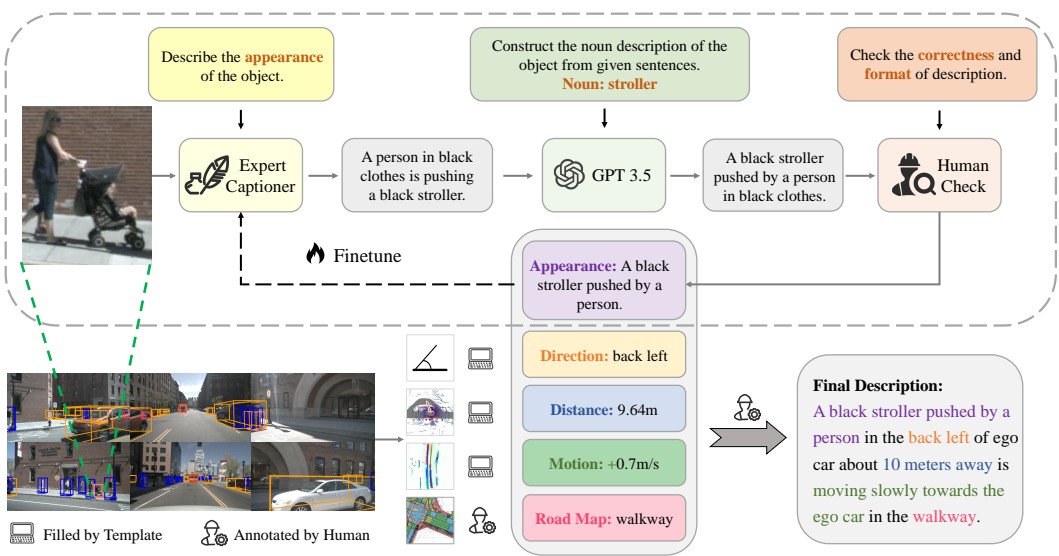

Figure 3: **The Annotation Process of nuDESIGN.** The entire process is divided into five sections, corresponding to the five components of the caption. Generally there are three types of annotation: model annotation, template annotation and human annotation, each of which is important to the effectiveness and efficiency of the whole process.

## 3 TASK

We introduce the task of dense driving captioning. The input is a driving scene with multi-view camera images and/or point cloud. Our goal is to design an architecture that can jointly localize the 3D bounding boxes for the underlying instances and generate corresponding detailed descriptions.

## 4 DATASET

The DESIGN dataset, which is based on nuScenes, comprises about 34k samples from 850 driving scenarios. Each sample consists of 6 camera images and the corresponding LiDAR point cloud. We provide detailed descriptions for each object in each sample, with the aim of complete coverage of all active agents in the driving scenes. The description is composed of five different components, including (1) Appearance: the visual appearance and category of the object; (2) Direction: the relative position to the ego car; (3) Distance: the distance to ego car; (4) Motion: the speed and orientation (whether the object is moving towards the ego car) of the object; and (5) Road Map: the location on the road map. Examples could be found in AppendixA.4.

In general, we employ five expert human annotators to work for about 2000 hours. The total number of language descriptions is about 2300k, with an average of 67.4 descriptions per sample and 2705.9 descriptions per scene. The total vocabulary consists of about 2k words. We show the properties of our dataset in Fig. 2. The descriptions are complex and diverse, requiring the model designed for this dataset to be capable of open understanding capabilities and precise spatial awareness. We show the comparison of DESIGN with existing dense captioning datasets in Tab. 1. To the best of our

knowledge, DESIGN is the first large-scale outdoor 3D dense captioning dataset, which contributes to the development of open-set perception in the autonomous driving community.

The overall annotating pipeline is illustrated in Fig. 3. In the following sections, we elaborate the annotating process of the five components of DESIGN successively.

## 4.1 APPEARANCE

Each object in DESIGN has a description of its visual appearance, which could be color, material, attire, etc. We believe such description containing common sense can help the downstream decision making process of autonomous driving systems. For example, a *stroller* should be paid more attention by the decision algorithm while a *bird* can possibly be ignored.

The annotating pipeline consists of two stages: auto collection and human feedback. The auto collection process is executed with a pre-trained caption model (named expert captioner) and LLM. Specifically, we first project the pre-labeled 3D bounding box to 2D, which is then used to crop the camera image to an image patch that primarily consists of one object. The image patch is then passed as input to a pre-trained expert captioning model (e.g. LLaMa-Adapter (Zhang et al., 2023)) to generate the appearance of each object. Afterwards, we employ GPT3.5 (Ouyang et al., 2022) to perform a slight refinement of the generated sentences. Finally, We ask human annotators to check the correctness of the content and format of the descriptions. To make expert captioner more in line with our requirements, after annotating 20% and 50% scenes, we fine-tune the expert captioner with the cropped image patches and the annotated ground truth sentences for one epoch. This close loop annotating strategy speeds up the whole process effectively.

## 4.2 SPATIAL POSITION

The spatial position of an object plays an import role in the planning of autonomous driving system, which is hard to recognize by existing vision-language models. Thus we also provide the description of the spatial position for each object, including the direction and distance.

### 4.2.1 DIRECTION

We define the direction of an object $O$ as the viewing angle from ego car. The viewing angle $\theta$ represents the angle between $\overrightarrow{P_O P_O'}$ and the orientation of the vehicle $R_{ego}$, formulated as: $\theta = \cos^{-1} \frac{(P_O - P_O') \cdot R_{ego}}{\|(P_O - P_O')\|_2 \|R_{ego}\|_2}$, where $P_O$ and $P_{ego}$ represents the position of target object and ego car, respectively. The viewing angle is then described implicitly by front, back, front left, front right, back left, and back right by several angle thresholds.

### 4.2.2 DISTANCE

The distance of an object is calculated by the euclidean distance between the bounding box centers of the target and ego car, formulated as: $d = \|P_O - P_O'\|_2$

## 4.3 MOTION

The motion of an object entails the concurrent state of its speed and orientation, conventionally denoted as a vector. We exploit an implicit manner to describe the motion of the object, focusing on whether it is approaching toward the ego car and how fast it is. We use three kinds of speed captions: (1) not moving, (2) moving slowly (3) moving quickly; and two direction captions: (1) towards the ego car, (2) away from ego car. The speed and orientation captions are generated in an automatic manner. We first differentiate the trajectory with respect to time to obtain the velocity of the object: $V_O = \frac{\delta \text{trajectory}}{\delta t}$ where $\delta$trajectory represents the position change of the ego car in $\delta t$ duration. Then we use $\|V_O\|_2$ to define the moving speed of the object and use the angle $\alpha$ between $\overrightarrow{O_{ego} O}$ and the speed vector $V_O$ to define the moving orientation of the object.

### 4.4 ROAD MAP

Roads are divided into different sections to ensure different functions. The object's location on the road map can assist the identification of the potential risk. For instance, a person in the sidewalk is less likely to influence the driving behavior than a person in the driving lane. Thus we provide the annotation that describes where an object locates on the road map. The annotation process of Road Map is conduct by human annotator with an instruction of "Describe the location of the object on the road map".

## 5 METHOD

The primary challenge of dense driving captioning task is how to jointly localize each object while capturing its rich visual concepts. We propose an end-to-end framework with the input of camera images and LiDAR to address this task. Specifically, we first embed the visual inputs to a bird's-eye view (BEV) space, and leverage a detection head to generate a set of 3D object proposals. Then the BEV feature map and object proposals are passed as input to a caption module to generate the natural language sentences.

### 5.1 BEV FEATURES AND REGION PROPOSALS

We adopt the BEV feature as the representation of the driving scenario. Given a sample with $N_{view}$ images and point cloud $P$, we exploit off-the-shelf methods (Liu et al., 2023; Li et al., 2022) to embed them into BEV space. Specifically, we first feed multi-camera images to a visual backbone (e.g., ResNet-101 (He et al., 2016)). A group of learnable BEV queries $Q \in \mathbb{R}^{H \times W \times C}$ is then employed to query features in BEV space from multi-camera features via attention mechanisms, where each query $Q_p \in \mathbb{R}^{1 \times C}$ located at $p = (x, y)$ of $Q$ is responsible for the corresponding grid cell region in the BEV plane.

Subsequently, we exploit a proposal module that takes the BEV feature as input to generate the object proposals $P_{0,1,...,N-1}$, where $N$ is the preset number of object queries. The process of proposal generation aligns with that in conventional detection head like DETR (Carion et al., 2020). The object proposals serve as position priors for the subsequent caption module.

### 5.2 CAPTION GENERATION MODULE

Following the proposal module, the BEV features and object proposals are passed as input to the caption generation module. As illustrated in Fig. 4, the caption generation module is comprised of a query transformer and a language decoder.

**Query transformer.** The query transformer extracts context-aware feature for each single object. We first tokenize the feature map $B_t \in \mathbb{R}^{H \times W \times C}$ along the channel dimension, resulting in $H \times W$ BEV tokens of size $C$. Then we create object queries by encoding the object proposals $D_{0,1,...,N-1}$ with a learnable MLP, resulting in object tokens with size of $N \times C$. The object tokens and BEV tokens are fed into a transformer (namely query transformer), where the object tokens and BEV tokens interact with each other through self-attention layers. The object queries are then input to a language decoder to generate language sentences.

**Language decoder.** We employ a frozen LLM as our language generator, which takes object queries as input and output descriptions for each object. To ensure the dimension consistency between object queries and LLM layers, we first use an MLP to transform the dimension of query embeddings. To bridge the modality gap, we employ an adapter (Zhang et al., 2023) to align the object representation with language modeling. The adapted object features serve as soft visual prompts that condition the LLM on the object representation, which are consumed to generate natural language sentences.

Considering the memory burden and optimization difficulty when generating hundreds of sentences during training, we do not input all the object queries to the language decoder. Instead, we filter the queries by a 3D hungarian assigner (Wang et al., 2022) and sample $N_s$ positive proposals randomly during training. During inference, we apply NMS (non-maximum suppression) to suppress overlapping proposals.

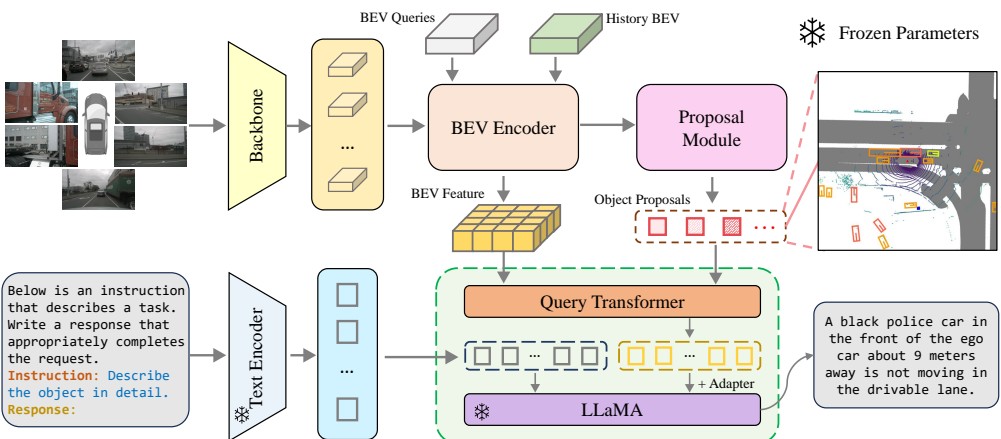

Figure 4: **Architecture of our proposed DESIGN-former.** We first embed the visual inputs to a bird's-eye view (BEV) space, and leverage a detection head to generate a set of 3D object proposals. Then the BEV feature map and object proposals are passed as input to a caption module to generate the natural language sentences.

## 5.3 TRAINING STRATEGY

To simplify the optimization of the network, we employ multiple pre-training approaches before the training of the entire model. Firstly, the BEV feature extractor and the proposal module is pre-trained within an object detection model. Then the wights of the bev feature extractor and proposal module are frozen and their results are utilized to train the caption generation module. Finally, the entire module is finetuned with a small learning rate.

## 5.4 LOSS FUNCTION

We employ the detection head proposed by Li et al. (2022), which utilizes $L_1$ loss to supervise 3D bounding box regression, and we optimize the cross-entropy loss for language generation. The final loss is defined as the weighted combination of them:

$$\mathcal{L} = \alpha \mathcal{L}_{det} + \beta \mathcal{L}_{des} \tag{1}$$

where we set $\alpha = 10$, $\beta = 1$ respectively in our experiments.

## 6 EXPERIMENTS

We perform a comprehensive evaluation of baseline models on the introduced dataset. The experiments are conducted independently on two different settings: (1) 3D dense captioning that takes multi-view camera images and/or point cloud as input and generates 3D bounding boxes and captions; (2) 2D dense captioning that takes single image as input and generates 2D bounding boxes and captions. Intuitively, 3D captioning has more advantages in spatial awareness, while 2D captioning outperforms in visual recognition.

## 6.1 METRICS

We leverage m@kIoU (Chen et al., 2021b) as the evaluating metrics on the dense driving captioning task, formulated as:

$$m@k\text{IoU} = \frac{1}{N} \sum_{i=0}^{N} m_i u_i \tag{2}$$

where $u_i \in \{0, 1\}$ is set to 1 if the IoU score for the $i^t h$ box is greater than k, otherwise 0 and $m$ represents the standard image captioning methods, including BLEU (Papineni et al., 2002), METEOR (Banerjee & Lavie, 2005), Rouge (Lin, 2004) and CIDEr (Vedantam et al., 2015), abbreviated as B, M, R, C, respectively.

| Method | Domain | Input | C@0.25 | B-4@0.25 | M@0.25 | R@0.25 | C@0.5 | B-4@0.5 | M@0.5 | R@0.5 |
|---|---|---|---|---|---|---|---|---|---|---|
| Ours-tiny | Outdoor | C | 158.7 | 51.2 | 38.4 | 73.1 | 160.2 | 51.4 | 38.5 | 73.3 |
| Ours-small | Outdoor | C | 190.0 | 57.6 | 41.1 | 77.3 | 192.5 | 57.8 | 41.3 | 77.6 |
| Ours-base | Outdoor | C | 190.3 | 57.5 | 41.4 | 77.6 | 192.2 | 57.7 | 41.5 | 77.7 |
| Scan2Cap (Chen et al., 2021b) | Indoor | L | 6.5 | 5.3 | 4.0 | 9.9 | 7.3 | 5.3 | 4.1 | 9.7 |
| X-Trans2Cap (Yuan et al., 2022) | Indoor | L | 114.5 | 45.0 | 34.4 | 69.0 | 118.0 | 45.3 | 34.6 | 69.3 |
| Ours | Outdoor | L | 171.1 | 53.1 | 38.3 | 72.0 | 172.8 | 54.2 | 38.7 | 74.2 |
| Scan2Cap (Chen et al., 2021b) | Indoor | C+L | 60.6 | 45.1 | 34.9 | 70.8 | 62.5 | 45.1 | 35.8 | 71.9 |
| X-Trans2Cap (Yuan et al., 2022) | Indoor | C+L | 133.0 | 45.9 | 35.5 | 71.8 | 141.9 | 46.7 | 35.7 | 72.6 |
| Vote2Cap-DETR (Chen et al., 2023b) | Indoor | C+L | 198.3 | 58.0 | 51.9 | 75.6 | 200.1 | 58.4 | 42.1 | 76.1 |
| Ours | outdoor | C+L | **218.0** | **60.2** | **43.8** | **76.2** | **220.3** | **61.5** | **50.9** | **80.1** |

Table 2: Comparison of 3D dense captioning results. Prior arts (Chen et al., 2021b; Yuan et al., 2022) are mainly designed for indoor scenes thus suboptimal for our setting, but they are trained to full convergence on nuDESIGN for fair comparison. Our proposed outdoor method outperforms them with a remarkable margin. Note that the captioning baselines are fine-tuned on the training set of nuDESIGN, instead of directly inferred with models trained on ScanNet.

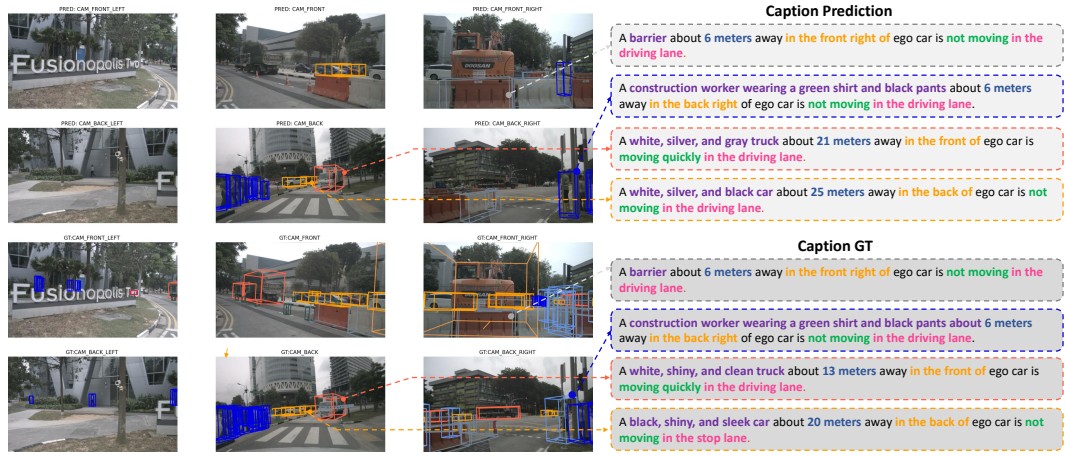

Figure 5: Qualitative results for our proposed DESIGN-former. We show the rendered 3D bounding boxes and corresponding captions. The captions are shown partially because of the page limitation.

## 6.2 3D DENSE CAPTIONING

We show results separately for different input modalities, where we evaluate with (1) multi-view camera images, (2) LiDAR point cloud and (3) both images and point cloud. Depending on the input modality, we exploit different BEV feature extractors: BEVFormer (Li et al., 2022), Centerpoint (Yin et al., 2021) and BEVFusion (Liu et al., 2023).

We benchmark existing methods for 3D dense captioning methods (Chen et al., 2021b; Yuan et al., 2022), most of which are initially proposed to address dense captioning task for 3D indoor scenes. The quantitative results are shown in Table. 2. We can see that out proposed method DESIGN-former gets better performance that previous works, demonstrating the effectiveness of our proposed model. And we notice that the Scan2Cap fails to complete the dense captioning task, which illustrates the difficulty of inferring outdoor object captions from LiDAR point cloud for such indoor methods. In order to verify whether the size of BEV has an impact on the model effect, we experiment on differ-ence BEV size, shown in the first three rows. We can see that the increased BEV size contributes to final performance.

## 6.3 2D DENSE CAPTIONING

Despite the superior spatial awareness of BEV representation, it compresses a lot of information when extracting features, like the height or geometry. Therefore, we also benchmark DESIGN2D with state-of-the-art 2D dense captioning methods (Johnson et al., 2016; Wu et al., 2022) as well

as our proposed 2D-DESIGN-former. The only difference of 2D-DESIGN-former from 3D is that the 2D version takes the CLIP features of image as input (instead of BEV features) and output 2D bounding boxes and corresponding captions. The results are shown in Table. 3. We can see that our method still outperform 2D approaches, demonstrating the effectiveness of DESIGN-former on 2D dense captioning. Note that the IOU in Tab. 3 is calculated through 2d bounding box, which is different from that in Tab. 2.

| Method | Input | C@0.25(2D) | B-4@0.25(2D) | M@0.25(2D) | R@0.25(2D) | C@0.5(2D) | B-4@0.5(2D) | M@0.5(2D) | R@0.5(2D) |
|---|---|---|---|---|---|---|---|---|---|
| DenseCap (Johnson et al., 2016) | C | 186.3 | 41.5 | 32.1 | 63.5 | 233.2 | 46.6 | 35.3 | 68.2 |
| GRITWu et al., 2022 | C | 216.5 | 63.8 | 44.3 | 81.9 | 264.5 | 67.1 | 46.2 | 84.0 |
| Ours | C | **220.0** | **64.3** | **44.7** | **82.2** | **274.6** | **67.4** | **46.7** | **84.4** |

Table 3: Comparison of 2D dense captioning results. The IOU threshold are calculated by 2D bounding boxes.

## 6.4 QUALITATIVE ANALYSIS

We show some qualitative results in Fig. 5, including the detection results and corresponding descriptions. DESIGN-former accurately recognizes most objects and provides sound descriptions. We observe that DESIGN-former is able to recognize the rare objects (or long-tailed objects), like the construction worker, which demonstrates the superior open scene understanding ability of DESIGN-former. More results can be found in the AppendixA.4.

## 6.5 OTHER ANALYSIS

A major concern of DESIGN is whether dense captions are sufficient to display the driving scenario for human drivers or Large Language Models. To test the contextual bridging ability of DESIGN, we conduct experiment Specifically, we utilize the driving agent (Fu et al., 2023) and conduct experiments by passing the captions to the driver agent to get the driving suggestions. The suggestions are then input to traditional autonomous driving systems, like UniAD (Hu et al., 2023), where we simply replace the learnable planning queries by the CLIP embedding (Radford et al., 2021) of the suggestions, results shown in Tab. 4. This demonstrates the potential of DESIGN as an intermediate representation that bridges autonomous driving systems and language foundation models.

We conduct another experiment that directly attaches metadata, expressed in real numbers describing geometric entities such as [-2.60m / s, 1.64m / s] or [-23.78m, 0.05m], to the caption. This would greatly challenge LLM since they are not natural language. We conduct experiments with this new version of captioning, where we attach meta data to the caption. The downstream planning modules follow the same protocol. This representation under-performs DESIGN as shown in Fig. 4. This experiment demonstrates that the performance gain brought by DESIGN is non-trivial and serves as another evidence of DESIGN's effectiveness.

We provide the prompt of the LLM driver agent and planning results visualization in Appendix.A.5.

| Method | Input | L2 1s | L2 2s | L2 3s | Avg. | Collision 1s | Collision 2s | Collision 3s | Avg. |
|---|---|---|---|---|---|---|---|---|---|
| UniAD (Hu et al., 2023) | 2D | **0.48** | **0.96** | 1.65 | **1.03** | 0.05 | 0.17 | 0.71 | 0.31 |
| Ours w/ metadata | 2D | 0.68 | 1.14 | 1.84 | 1.19 | 0.07 | 0.15 | 0.61 | 0.28 |
| Ours | 2D | 0.53 | 0.98 | **1.64** | 1.05 | **0.03** | **0.12** | **0.55** | **0.23** |

Table 4: Comparison of planning results on nuScenes. With language suggestions, the results of UniAD have been greatly improved, especially on collision rate despite the l2 loss has been slightly improved.

## 7 CONCLUSIONS

We introduce the novel task of 3D dense beyond nouns, abbreviated as DESIGN, aiming to provide a middleware to bridge the perception module of autonomous driving systems and language foundation models. To support the development of this task, we collect the nuDESIGN dataset, which contains 2300k descriptions for 1165k objects in 850 scenes. We propose an end-to-end method for jointly localizing an object and generating corresponding descriptions. Experiments on down-

streaming planning tasks verify the effectiveness of this new middleware representation on bridging AD systems and foundation models.

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

## A   APPENDIX

### A.1   OVERVIEW

Section. A.1 Overview of Appendix.

Section. A.2 Driving QA Module.

Section. A.3 More Dataset Details.

Section. A.4 Visualization of Caption and Diversity.

Section. A.5 LLM Driver Agent.

### A.2   2D QUESTION ANSWERING

**Dataset**    Based on DESIGN, we further contribute a visual question answering (VQA) benchmark in autonomous driving scenario regarded as driving QA. According to previous caption templates, we designed various kinds of question with structured data generated from dense captioning of nuscenes, encompassing existence, counting, query-object, query-status, and comparison, inspired by CLEVR benchmark (Johnson et al., 2017). For instance, the template for questions of the existence type can be expressed as follows: "Are there any $< A2 > < O2 >$ to the $< R >$ of the $< A1 > < O1 >$?," where $< A2 >$, $< O2 >$, and $< R >$ correspond to distinct parameter types, specifically attributes, objects, and relationships. Eventually, we obtained a total of 468K question-answer pairs on 840 scenes from the annotated nuScenes training and validation split, with 385K pairs for training and 83K for testing.

**Experiment**    Furthermore, We perform automatic answer generation by utilizing a pre-trained VLM(e.g. LLaMa (Touvron et al., 2023a)), subsequently fine-tuning it to combine previous questions and multi-view camera image features of individual LiDAR samples, thus producing logically accurate and reasonable answering results. After that, we establish a baseline model and extensively evaluated the performance of existing method for visual question answering task on driving QA. To assess the performance of the QA task, we employ Top-1 accuracy as our evaluation metric, in line with the common practice in various VQA studies (Antol et al., 2015). Additionally, we incorporated sentence evaluation metrics commonly utilized in image captioning models since certain questions had multiple valid answer expressions. Specifically, we integrated BLEU (Papineni et al., 2002), ROUGE (Lin, 2004), and METEOR (Banerjee & Lavie, 2005) metrics to evaluate the answer matching. We compared our method trained with single view and multiview image features. The results are shown in Table. 5.Our proposed baseline with multi-view BEV image features outperforms single view trained model with a clear margin.

**Visualization Results**    To gain deeper insights into the challenges posed by the driving QA dataset and to validate the efficacy of our proposed baseline models, we have chosen a selection of samples

| Model | Accuracy | BLEU-1 | BLEU-2 | BLEU-3 | BLEU-4 | ROUGE | METEOR |
|---|---|---|---|---|---|---|---|
| Ours | 46.02 | 56.67 | 19.38 | **0.20** | 0.02 | 51.14 | 56.68 |
| Ours(+multiview) | **47.77** | **58.45** | **19.50** | 0.19 | **0.02** | **52.97** | **58.46** |

Table 5: Performance comparison of 2D question answering results.

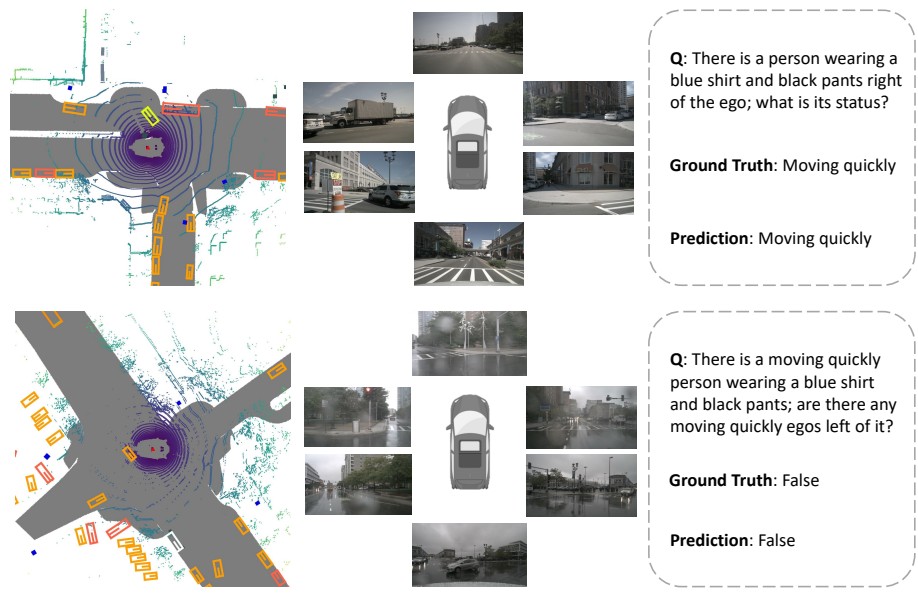

Figure 6: Qualitative results of driving QA.

from the test split for visualization, as depicted in Fig. 6 and Fig. 7. These visualizations include point clouds, surround-view images, questions, ground truth answers, and the predictions generated by our baseline models.

## A.3 DATASET DETAILS

**Statistics of word frequency.** In Fig. 8, we show the top-200 word frequency, whose frequency represents the logarithmic percentage of each word. Visualization implies that The top 100 words (e.g. moving, black, right), which are used to describe common scenes account for more than 90% of the total vocabulary in the dataset, while other words only appear in specific scenes.

**Statistics of sentence length frequency.** We provide the statistics of the length of all caption sentences in Fig. 9. The sentences length ranges from 15 to 28 and the frequency represents the logarithmic percentage of each length. It can be seen that the number of words in most caption sentences is 15 to 25.

**Visualization of objects' spatial position** Given the object's viewing angle and distance from the ego car, the projection of the distance in the ego car direction and perpendicular to the ego car direction can be calculated. Following this, we visualize the distribution of objects' spatial position separately according to their category, as shown in Fig. 10 where (a) - (e) represents the position distribution of animals, humans, movable objects, static objects, and vehicles relative to the ego car respectively. Besides, the visualization of the mean distribution of each kind of object is shown in Fig. 10 (f), which demonstrates the even distribution of all kinds except the ambulance. (This is because ambulances rarely appear in the dataset.)

## A.4 VISUALIZATION

**Visualization of predicted caption and ground truth.** As shown in 11, we demonstrate the effectiveness of our method. Specifically, we showcase common and long-tailed objects in four categories of road scenes, including cars, trucks, construction worker, and traffic cones. Our method exhibits

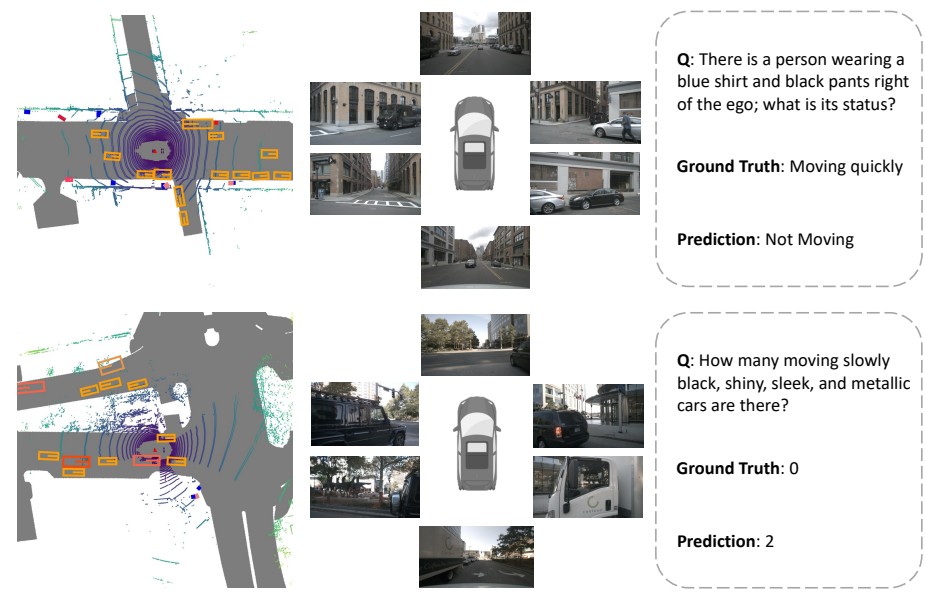

Figure 7: Bad cases of driving QA.

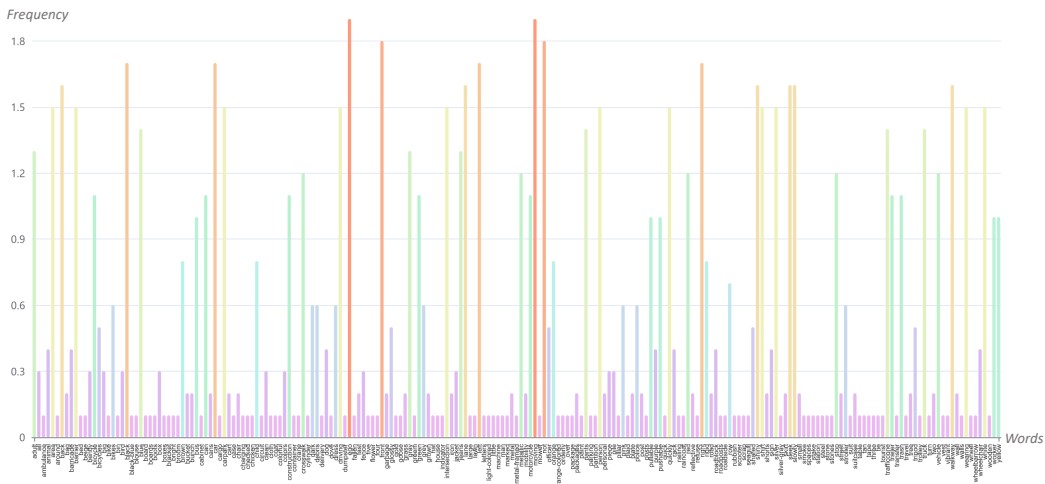

Figure 8: Statistics of word frequency.

only minor prediction errors in terms of distance and object appearance, such as color (obj.2 and obj.4). And it accurately predicts the motion states and traffic positions.

**Visualization of long-tailed objects**   In 12 and 13, we showcase some long-tail objects in our dataset along with their corresponding ground truth caption, which demonstrates the richness and diversity of our dataset.

## A.5   LLMDRIVERAGENT

In order to explore the effectiveness of large language model (LLM) on autonomous vehicle planning and decision, following Fu et al. (2023), we generate driving prompts as shown in Fig. 14. These prompts are then fed into ChatGPT to obtain autonomous driving instructions, creating what we call an LLMDriverAgent. We use these the output of LLMDriverAgent, or say driving instruction, shown in Fig. 18, as additional inputs to UniAD (the framework is shown in Fig 17) The results,

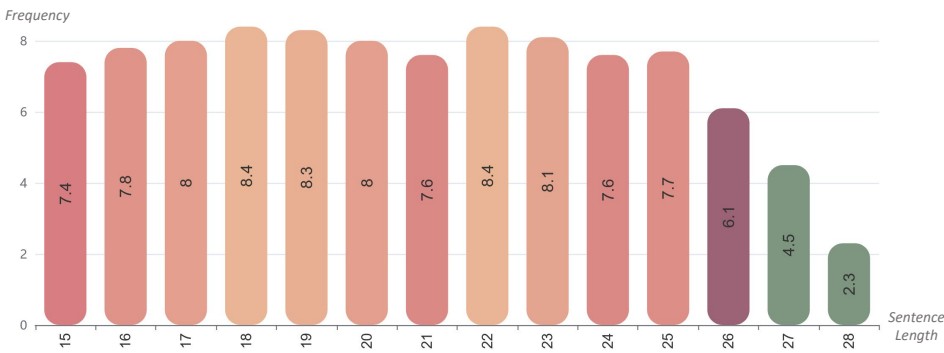

Figure 9: Statistics of sentence length frequency.

shown in Fig. 15, demonstrate the effectiveness of natural language commands on autonomous vehicle planning.

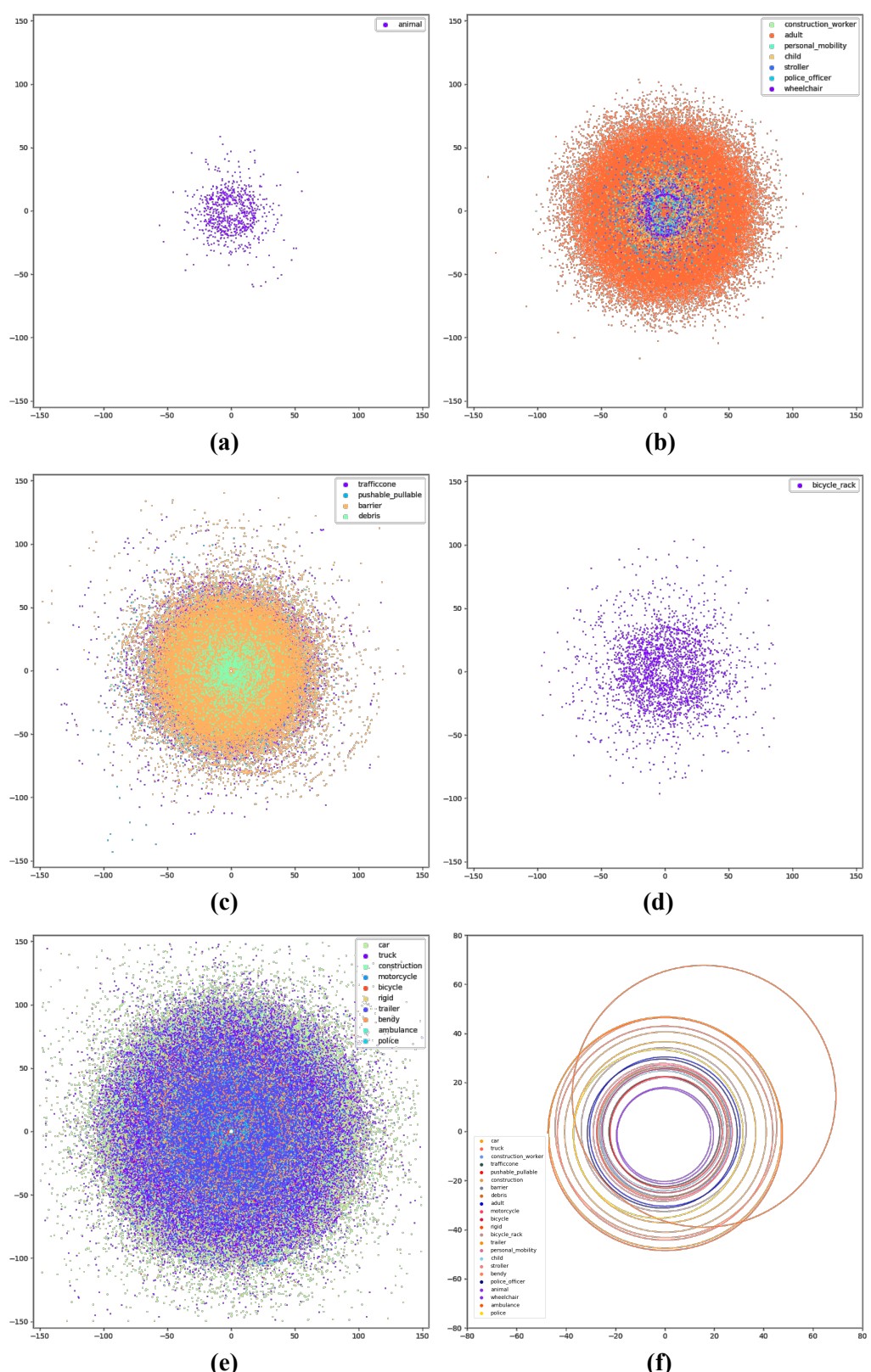

Figure 10: Visualization of objects' spatial position. (a) - (e) represents the distribution of objects' position. (f) shows the mean distribution of each kind's position.

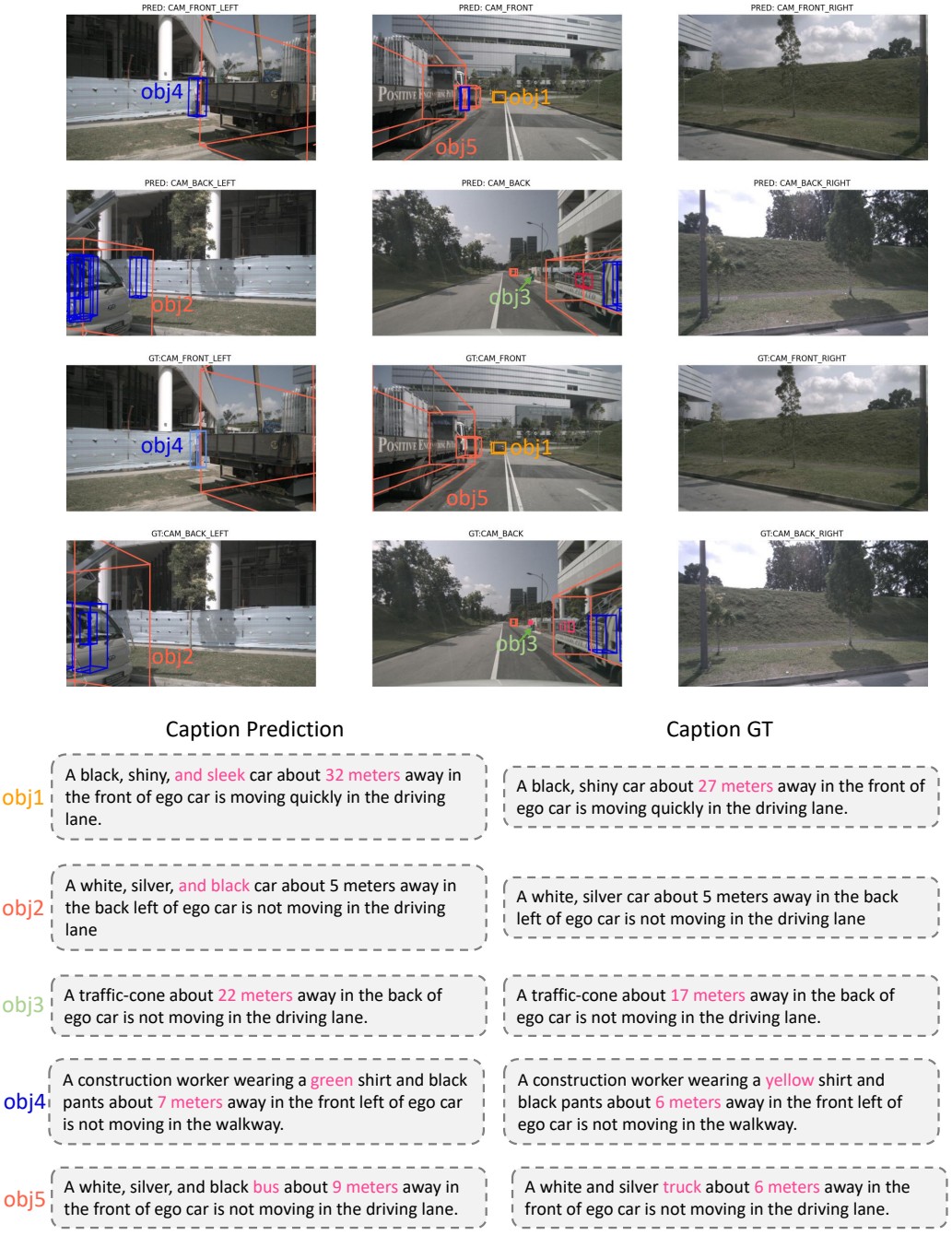

Figure 11: Visualization of predicted caption and ground truth. We align the objects and mark the difference between prediction and ground truth with pink color.

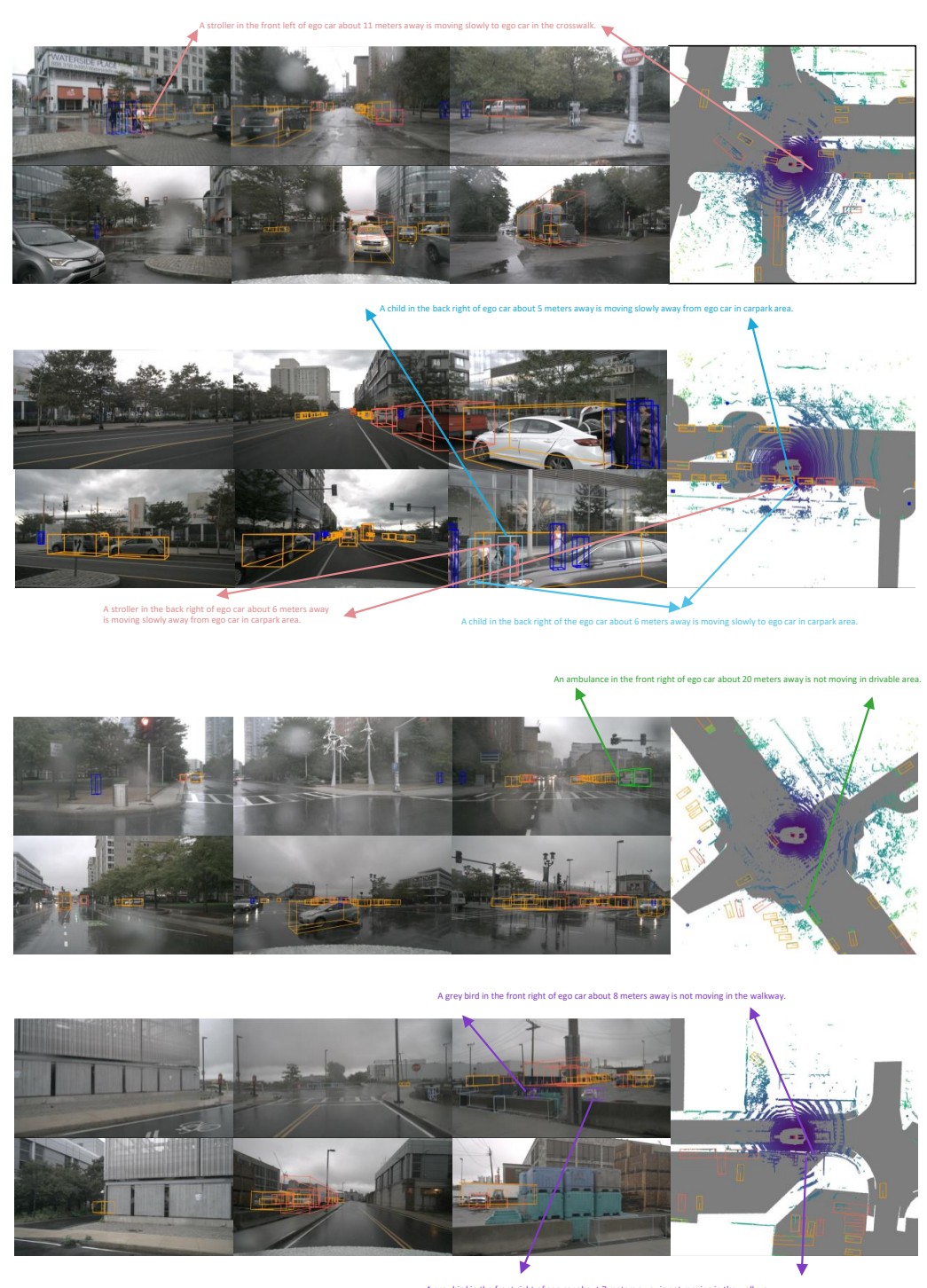

Figure 12: Visualization of long-tailed objects.

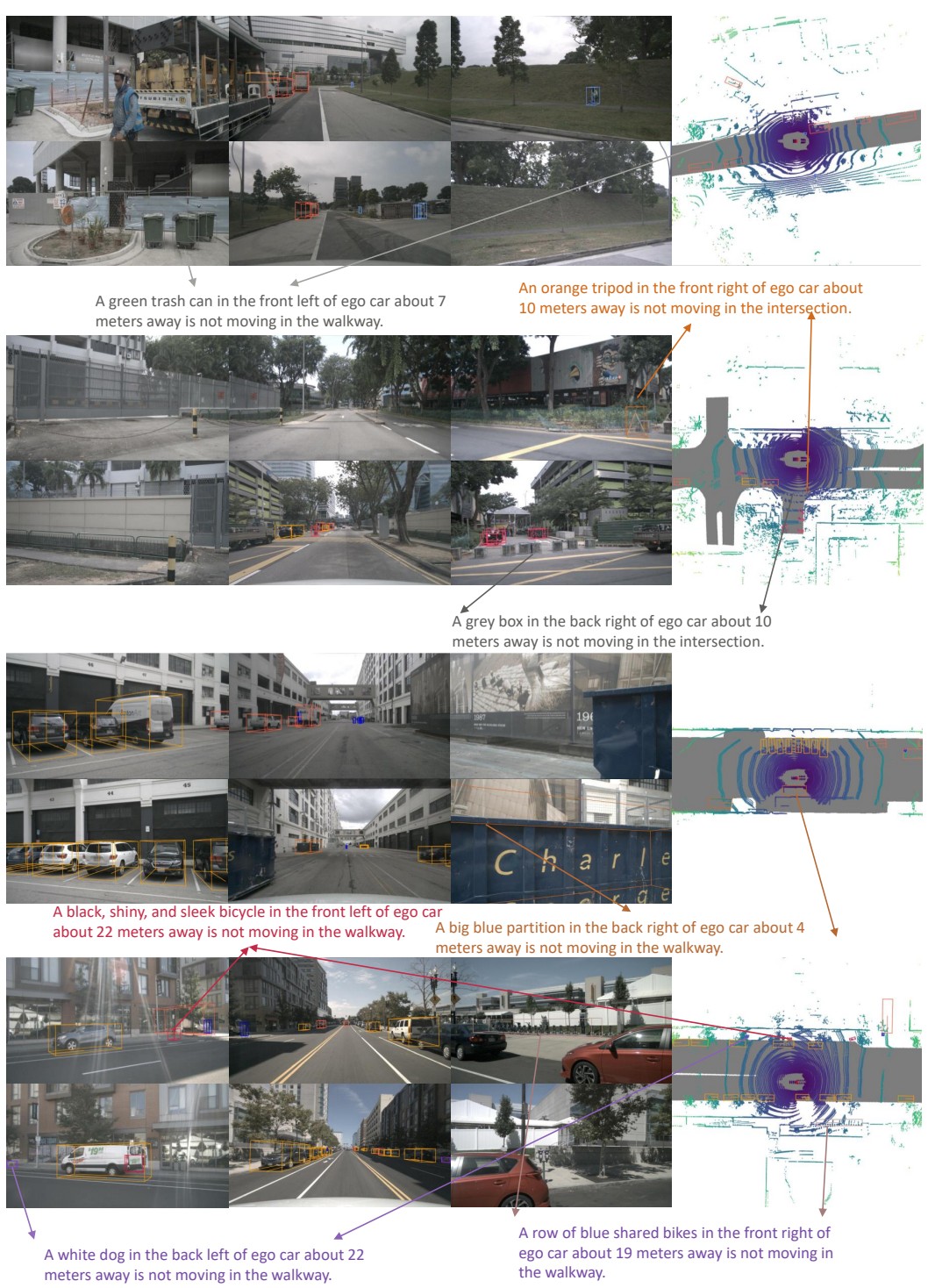

Figure 13: Visualization of long-tailed objects.

## Prompt

**Part I**

### INFORMATION

I will provide the speed of the vehicle you are driving (ego car) and the position and speed of other surrounding objects that may affect your driving behavior.
ego_car_velocity: The current speed of the car you are driving
object_relative_position: The position of objects in the scene relative to the ego car.
    in the back left of N meters away: The object is situated in the left rear adjacent lane of the ego car about N meters away.
    in the back right of N meters away: The object is situated in the right rear adjacent lane of the ego car about N meters away.
    in the back of N meters away: The object is situated in the rear same lane of the ego car about N meters away.
    in the front left of N meters away: The object is situated in the left front adjacent lane of the ego car about N meters away.
    in the front right of N meters away: The object is situated in the right front adjacent lane of the ego car about N meters away.
    in the front of N meters away: The object is situated in the front same lane of the ego car about N meters away.
motion_caption: The current state of motion of the object.
car_direction: The current lane option that the local planner is guiding the ego car towards. You can think of it as the current action. car_direction contains three values: LEFT, RIGHT, FORWARD.
    LEFT: The ego car is about to make a left turn and change lanes to the left adjacent lane.
    RIGHT: The ego car is about to make a right turn and change lanes to the right adjacent lane.
    FORWARD: The ego car will continue driving on the current lane, possibly accelerating, decelerating, or maintaining its current state.

SAFETY CRITERIA:
1. If there are no stopped vehicles in the front left of the adjacent lane, and the vehicle behind the left adjacent lane is moving slowly or stopped, left lane change is allowed.
2. If there are no stopped vehicles in the front right of the adjacent lane, and the vehicle behind the right adjacent lane is moving slowly or stopped, right lane change is allowed.
3. If there is a stationary object on the same lane ahead, the ego car can choose to move to an adjacent lane.
4. If there is a fast-moving vehicle within 25m behind the left adjacent lane, it is very dangerous to change lanes to the left.
5. If there is a fast-moving vehicle within 25m behind the right adjacent lane, it is very dangerous to change lanes to the right.

**Part II**

### ACTION LIBRARY

These are pre-coded actions that the ego car can directly implement.
1. ACTION_NAME: STOPDescription: This function performs an emergency stop. Steering remains unchanged to keep the ego car in its lane.
2. ACTION_NAME: CAHNGE_LEFT_TO_ADJACENT_LANEDescription: This function makes the ego car change to the left of the adjacent lane.
3. ACTION_NAME: CHANGE_RIGHT_TO_ADJACENT_LANEDescription: This function makes the ego car change to the right of the adjacent lane.
All of the below functions(4-7) perform lane-keeping driving, meaning that when these functions are executed, you will continue along the lane you are currently traveling on.
4. ACTION_NAME: KEEP_DRIVING_AT_CURRENT_SPEEDDescription: This function maintains the current speed of the ego car on the lane it is currently traveling on.
5. ACTION_NAME: KEEP_CURRENT_STATUSDrive according to the state specified by the Car_Direction without changing the current state.
6. ACTION_NAME: SPEED_UPDescription: This function accelerates the ego car by 0.5 km/h, and continues driving in the current direction and lane.
7. ACTION_NAME: SPEED_DOWNDescription: This function decelerates the ego car by 0.5 km/h, down to a minimum of zero, and continues driving in the current direction and lane.

**Part III**

### DRIVING BEHAVIOR

Perform three tasks: Situation Understanding, Reasoning, and Action Commanding. Use the provided INFORMATION to guide the ego car's next action.
1. Situation Understanding:
    1) If the car behind the adjacent lane is driving very fast, it is often impossible to change lanes onto this lane.
    2) Analyze and describe the ego car's situation using all of the given INFORMATION, and mention all the parts of INFORMATION
        you used in the Situation.
    3) Think step by step.
    4) Describe as concisely as possible and keep the content of this part within 30 words.
2. Reasoning:
    1) Decide the ego car's next action based on Situation Understanding.
    2) Prioritize safety for both drivers and pedestrians.
    3) Reach the destination quickly, ensuring safety.
    4) Give clear, detailed instructions for the action.
    5) Ensure the continuity of driving operations.
    6) Think step by step.
    7) Describe as concisely as possible and keep the content of this part within 30 words.
3. Action Commanding:
    1) Specify the action using the ACTION LIBRARY format.
    2) Follow this structure: "Action Commanding": "ACTION_NAME".
    3) Use a single phrase for each action. Avoid multiple actions or extra content.

**Part IV**

### EXAMPLE

**Situation Understanding:**
    The ego car's ego_car_velocity is 3.32 m/s, and Car_Direction is 'RIGHT'. White car 33332's object_relative_position is 'in the front of ego car 10 meters away', which motion_caption is 'moving slowly'. Black, shiny, and sleek car 24163's object_relative_position is 'in the back left of ego car 18 meters away', which motion_caption is 'not moving'. Black car 33456's object_relative_position is 'in the back right of ego car 9 meters away', which motion_caption is 'moving quickly'.
**Reasoning:**
    The Car_Direction of the ego car is RIGHT, but there is a fast car 9m away from the right lane behind it. It may be dangerous if you continue to change to the right.
**Action Commanding:**
    STOP.

**Situation Understanding:**
    The ego car's ego_car_velocity is 6.80 m/s, and Car_Direction is 'FORWARD'. White car 11852's object_relative_position is 'in the front of ego car 15 meters away', which motion_caption is 'not moving'. Black, shiny, and sleek car 64163's object_relative_position is 'in the back right of ego car 18 meters away', which motion_caption is 'moving quickly'.
**Reasoning:**
    The Car_Direction of ego car is FORWARD. There is a stationary car 15 meters in front of it, and a car 18 meters behind and to the right is traveling faster, so turning left is a good choice.
**Action Commanding:**
    CAHNGE_LEFT_TO_ADJACENT_LANE.

**More...**

**Part V**

### FORMAT

You should only respond in the format as described below.
RESPONSE FORMAT: {"Situation Understanding": "......", "Reasoning": "......", "Action Commanding": "......"}.

Figure 14: Prompt input to LLM driver agent.

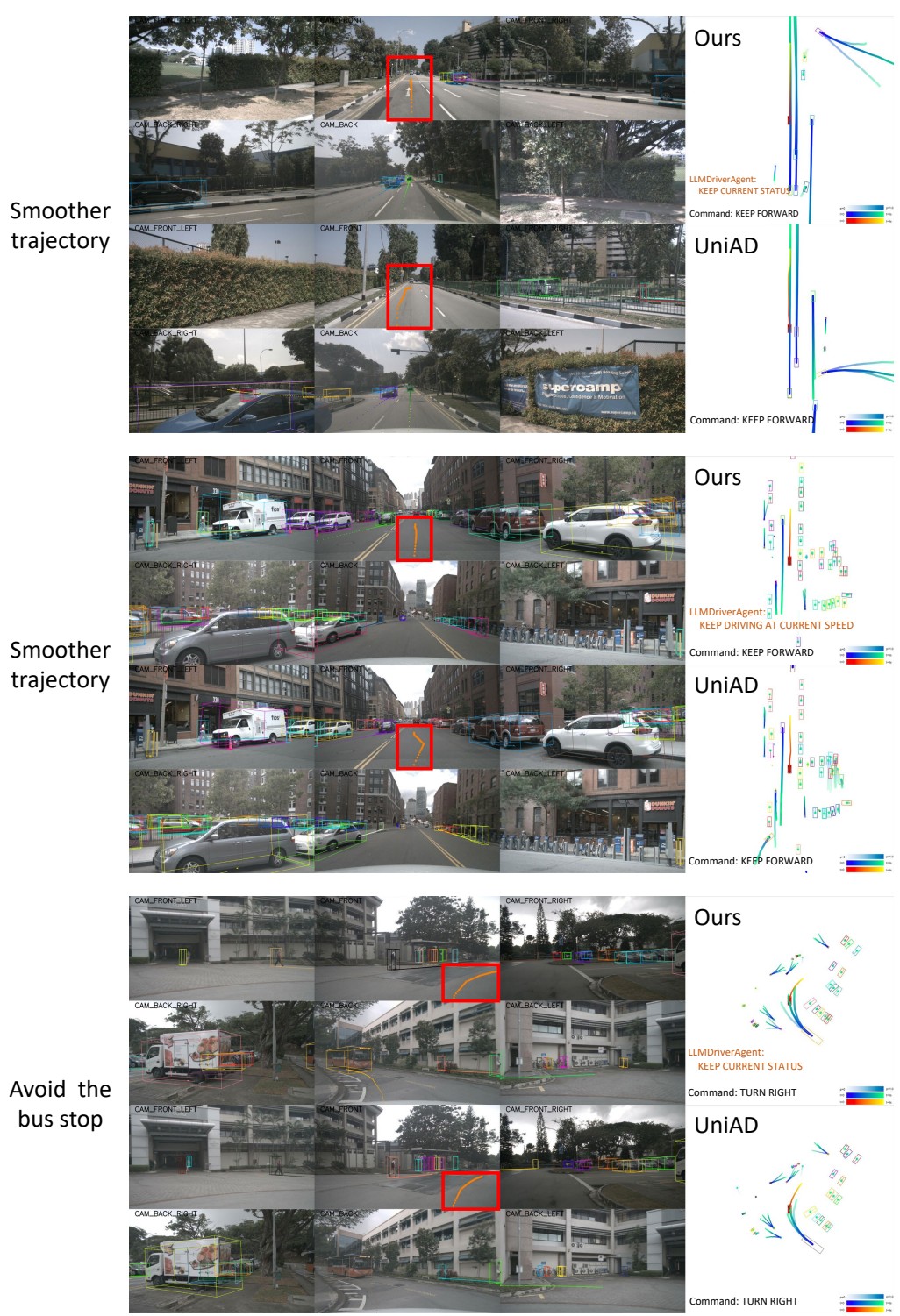

Figure 15: Qualitative results of UniAD and our proposed method. Red boxes show the comparison between UniAD's trajectories and ours. In case 1 and 2, our predicted trajectories are more smoother as our method is more aware of interaction with environment and vehicles. In case 3, our predicted trajectory successfully avoid the bus stop where UniAD fails.

```
"Situation Understanding":
    "The ego car's ego_car_velocity is 5.07 m/s and Car_Direction is
'FORWARD'. There is a slow-moving vehicle in the front of the ego car
and a fast-moving motorcycle in the front right of the ego car. There
is also a slow-moving vehicle in the back left of the ego car and a
stationary traffic cone in the back right of the ego car.",

"Reasoning":
    "Based on the current state of the ego car and the SAFETY CRITERIA
it is not safe to change lanes left or right. The ego car should
continue driving straight in the current lane and maintain a safe
distance from the slow-moving vehicle in the front.",

"Action Commanding":
    "KEEP_CURRENT_STATUS"
```

```
"Situation Understanding":
    "The ego car is currently traveling at a speed of 3.62 m/s and is
facing forward. There are several objects in the front and back of the
ego car including a black motorcycle a person wearing a white shirt
and black pants a person wearing a black shirt and blue jeans a black
car a silver car and a traffic cone. The motorcycle and the person
wearing a white shirt are moving quickly while the other objects are
moving slowly or not at all.",

"Reasoning":
    "Based on the current situation it is important to prioritize
safety and avoid any potential collisions. The ego car should slow
down to a safe speed and take appropriate action to avoid the objects
in its path.",

"Action Commanding":
    "SPEED_DOWN"
```

```
"Situation Understanding":
    "The ego car's ego_car_velocity is 4.87 m/s and car_direction is
'FORWARD'. There is a motorcycle and a person in the front of the ego
car and a car in the front right of the ego car. There are also cars
in the back of the ego car and in the back right of the ego car.",

"Reasoning":
    "Based on the current state it is necessary to change lanes right
to avoid the obstacles in the front and back of the ego car.",

"Action Commanding":
    "CHANGE_RIGHT_TO_ADJACENT_LANE"
```

```
"Situation Understanding":
    "The ego car's ego_car_velocity is 4.62 m/s and car_direction is
'LEFT'. Black shiny and sleek motorcycle 57922's
object_relative_position is 'in the front left of ego car 41 meters
away' which motion_caption is 'moving quickly' in the driving lane;",

"Reasoning":
    "There is a fast-moving vehicle in the left adjacent lane and the
ego car is traveling at a relatively slow speed. To ensure safety and
reach the destination quickly it is necessary to speed up.",

"Action Commanding":
    "SPEED_UP"
```

Figure 16: The example outputs of LLM driver agent.

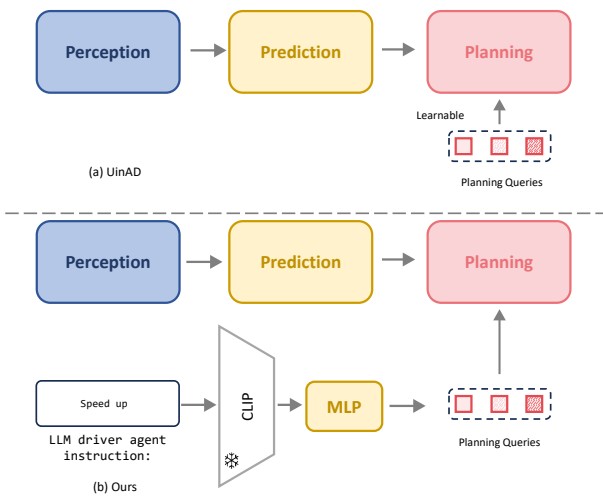

Figure 17: The architecture of UniAD + LLMDriverAgent in Tab. 4.

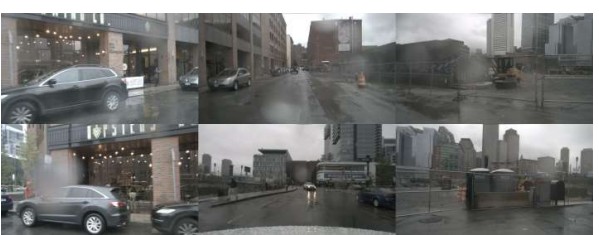

**"Situation Understanding":**
    "Ego car's velocity is 6.81 m/s, moving FORWARD. Obstacles include a slow-moving car 9m ahead in the left adjacent lane, a stationary traffic cone 18m directly ahead, and various objects (vehicles, a person, barriers) in close proximity in adjacent lanes and behind.",

**"Reasoning":**
    "With a stationary object ahead and slow-moving or stationary obstacles nearby in all lanes, it's safest to reduce speed to avoid collisions.",

**"Action Commanding":**
    "SPEED_DOWN"

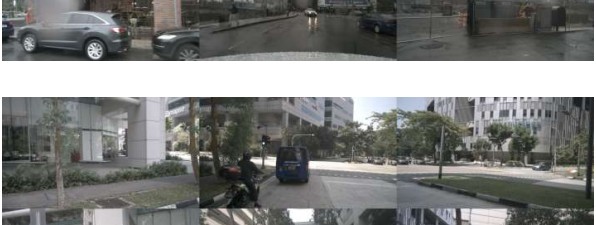

**"Situation Understanding":**
    "Ego car velocity is 0.25 m/s, barely moving FORWARD. A slow-moving motorcycle is 6m ahead, and a quickly moving blue car is 12m ahead in the driving lane. Traffic cones are 29m away, and a stationary blue car is 32m away in the front right. Behind, stationary cars are 5m and 13m away, and pedestrians are 14m and 15m away in the back left, moving slowly.",

**"Reasoning":**
    "With slow and fast-moving vehicles ahead and stationary traffic behind, increase speed slightly to maintain traffic flow.",

**"Action Commanding":**
    "SPEED_UP"

Figure 18: More example outputs of LLM driver agent.

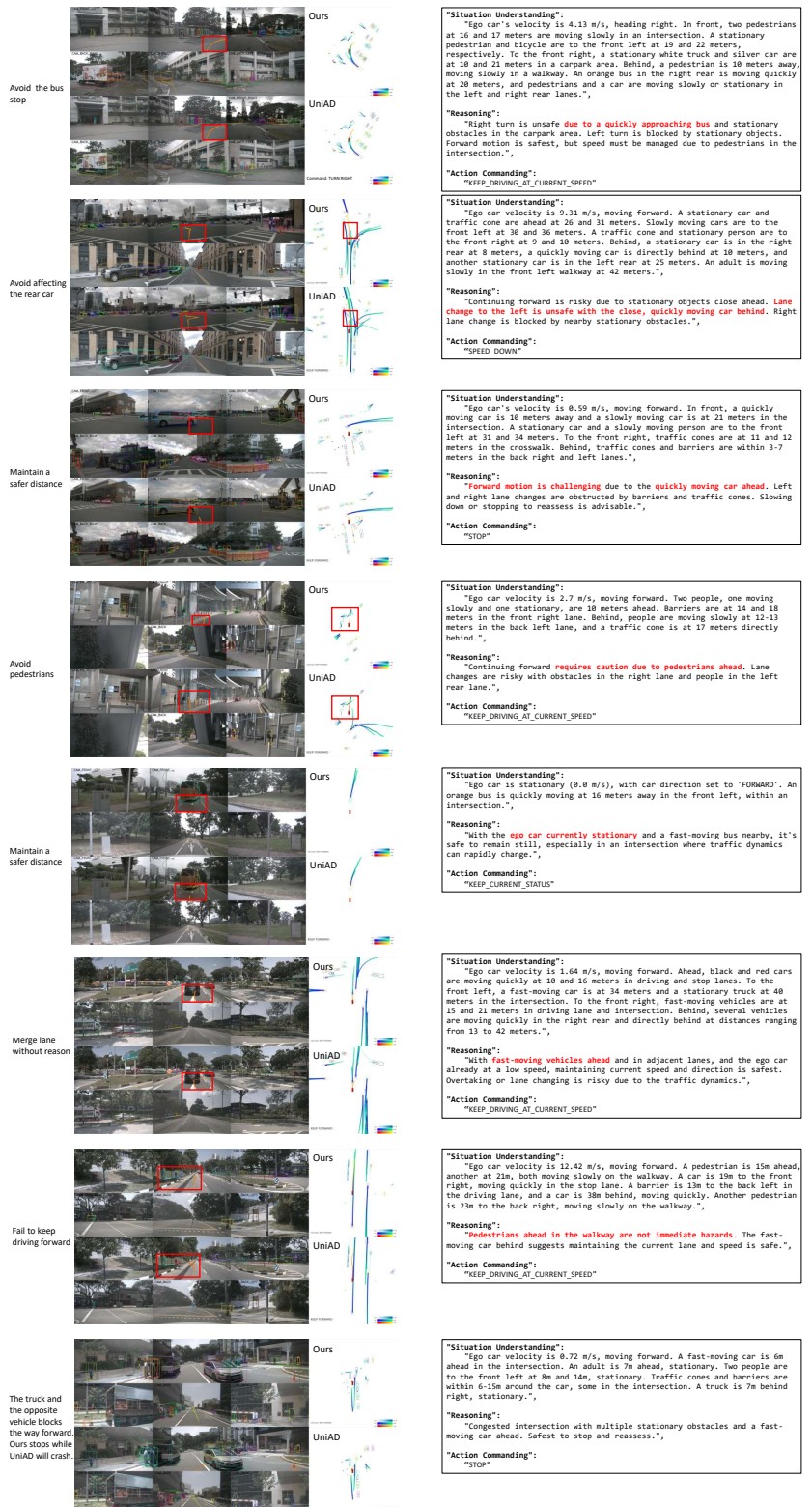

Figure 19: More qualitative results of UniAD and our proposed method. Red boxes show the comparison between UniAD's trajectories and ours. The right colume shows the output of LLM agent.

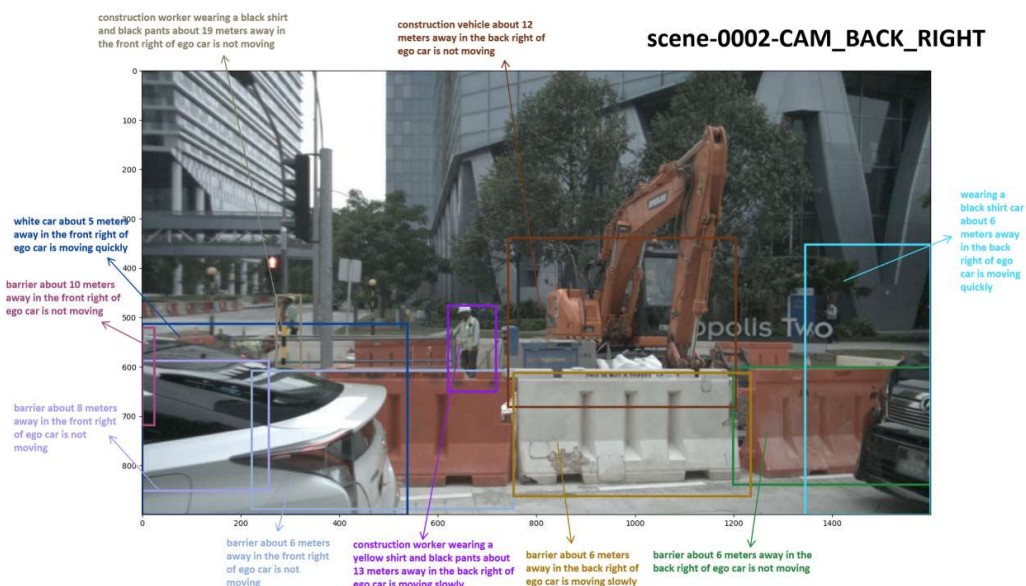

Figure 20: Visualization of 2D dense captioning.

