# OpenReview forum: "3D Dense Captioning beyond Nouns: A Middleware for Autonomous Driving"
_ICLR.cc/2024/Conference — Submitted to ICLR 2024_

### Official Review · Reviewer_R3EY · 2023-10-17

**Soundness:** 2 fair
**Presentation:** 2 fair
**Contribution:** 3 good
**Rating:** 6
**Confidence:** 5

**Summary:**

This paper introduces a new dense caption dataset for autonomous driving. They design baselines by combining BEV and LLM. Some exploration experiments are conducted to show the effectiveness of the dense caption.1. Using LLM for AD is a hot topic in recent days, which is intriguing
2. The promise of open sourcing a dataset with the cost of five expert human annotators to work for about 2000 hours is useful for the community.

**Strengths:**

1. Using LLM for AD is a hot topic in recent days, which is intriguing.
2. The promise of open sourcing a dataset with the cost of five expert human annotators to work for about 2000 hours could be helpful for the community.

**Weaknesses:**

The actual usefullness of such a dense caption dataset is still unclear since there could lots of complex obstacles or information should be noticed in the driving scene, which might not be easy to describe precisely in language or covered by the template. (It is note worthy that 3D occupancy and open-set scene understanding are hot topics discussed in the community to deal with such problems). How  the dense caption actually could help autonomous driving system is actually unclear.

Though in Table 6, the authors try to show that dense captions are useful. However, after applying such a complicated (and slow) pipeline to obtain the the CLIP embedding of the suggestions, it makes no significant difference compared with the original random initialized planning query in UniAD.

In summary, I appreciate the efforts the authors put to curate the dataset and I think it could be valuable for the further study, though it might not be clear how to properly use it right now. Thus, I give a borderline accept.

**Questions:**

See weakness part.

---

> ### Author Response · Authors · 2023-11-17
> **Response to reviewer R3EY**
>
> We would like to thank R3EY for generally positive feedback. Here we respond to two concerns:
>
> Q1: **usefullness of such a dense caption dataset**.
>
> We would like to remark on the effectiveness of the DESIGN middleware from three perspectives. **Firstly**, as demonstrated by the UniAD experiment in Table.4, DESIGN can significantly improve the collision rate by a relative margin of 25.8% (from 0.31 to 0.23), while achieving comparable L2 (from 1.03 to 1.05). We argue that collision rate is a more meaningful indicator than L2 for autonomous driving, so this result is significant enough to demonstrate the effectiveness of DESIGN. **Secondly**, we hypothesize (with all due respect) that R3EY may feel the UniAD experiment does not contain enough datapoints. In this regard, we provide another datapoint in Table.4 that directly turns the exact real numbers into natural language descriptions. This representation underperforms DESIGN. This datapoint demonstrates that the performance gain brought by DESIGN is non-trivial and serves as another evidence of DESIGN's effectiveness. **Thirdly**, we evaluate in another setting that shows DESIGN can also improve detection performance. The results below show that the DESIGN-former outperforms BEVFormer in mAP and NDS, suggesting that the cues from language helps to regularize the middle representation of BEV features and improve the performance of the bounding box detection. This serves as another evidence that DESIGN has some superiority over an existing middleware that is a subset of DESIGN (i.e., 3D bounding boxes).
>
> Q2: **difference compared with the original random initialized planning query**.
>
> We would like to clarify the differences between UniAD and our approach. We can see in the last example of Fig.19, a white truck in ego lane blocks the way. The UniAD tries to merge to the right lane and collides with the silver car on the right lane, while our driving model stops to avoid going against the flow of traffic and thus avoids the collision with right car. We think this introduction of reasoning ability of large language models to driving model remarks the difference between UniAD and our approach.

---

### Official Review · Reviewer_rzAd · 2023-10-30

**Soundness:** 2 fair
**Presentation:** 2 fair
**Contribution:** 2 fair
**Rating:** 3
**Confidence:** 4

**Summary:**

This paper targets the captioning task in the autonomous driving domain. For the dataset part, the authors use a LLaMa-Adapter and GPT-3.5 to describe the appearance of pre-labeled objects and generate spatial position-related descriptions with the ground truth from the original nuScenes dataset. For the methodology part, BEVFusion-MIT and BEVFormer are adopted to get the BEV feature and object proposals, which are then fused and sent to LLaMa via an adapter to predict the final captions. Experiments are conducted on 3D object proposals and 2D proposals. Besides, the authors also show that providing an End-to-end method (UniAD) with LLM-based planning queries can slightly improve the performance.

**Strengths:**

The paper has two parts, dataset and model. Different components, annotation processes, and model architecture are basically clearly written and easy to follow. The analysis of the captioning dataset is relatively comprehensive, including multiple graphs illustrating distributions of words and sentences. The authors also present a baseline method to realize the captioning task. The method builds upon the existing detection method and is easy to reproduce.

**Weaknesses:**

The reviewer is not fully convinced by the importance and soundness of the `comprehensiveness` of the proposed task and dataset.

- The annotation process starts with pre-labeled 3D/2D bounding boxes. The proposed task **does not go beyond previous definitions in terms of object-level comprehensiveness**. Why not directly attach more metadata to the original bounding box information?
- The authors claim that their description is beyond nouns compared to existing 3D bounding boxes. The current nuScenes dataset provides bounding boxes with position, heading angle, velocity, and category. Besides converting these annotations into language, the proposed dataset mainly gives more descriptions of objects' visual appearance. It can be seen in Fig. 2(a,b), that **most words are about objects' colors**, which are not so helpful for driving IMO. The dataset also **drops the heading angle and accurate velocity** information in classical box information which are sometimes critical for downstream tasks. In all, the new middleware, language, does not take more helpful information for driving.
- The over-claim of 'scalable automatic labeling pipeline': The annotation takes human evaluation in the loop, which is not fully automatic and thus not easy to scale. The influence of human check should be explained if a finetuned expert captioner can realize full automatic annotation when scaling to more diverse data.

Clarifications on experiments are needed.

- Comparisons to previous methods do not bring more insights. It is possible that the gains come from better detectors. What are the corresponding captions in 2D dense captioning?
- Though the method serves as a baseline for the proposed task, it is beneficial to have some ablations to provide more insights, eg., ablations on adapters or training strategies.

Some other points about writing:

- Sec 2.3 (Related work: Learning-based / End-to-end Autonomous Driving) seems not so relevant to the main contents of the paper. The reviewer suggests removing this part.
- The input modality in Tab. 1 may have some incorrect parts. For example, Talk2Car/Refer-KITTI builds on top of nuScenes/KITTI thus it can also take LiDAR as input.
- P_{ego} instead of P_{O} in Equation 1. Equations 1-3 are relatively too naive to show them in a single line IMO.

**Questions:**

- Are the object proposals boxes (N x 9) or features (N x C)?
- In Fig. 15, how to tell that UniAD fails to avoid the bus stop from the visualization?

---

> ### Author Response · Authors · 2023-11-17
> **Response to reviewer rzAd**
>
> We thank rzAd for these constructive feedbacks. Here we respond to ten concerns one by one:
>
> Q0: **slightly improve the performance**.
>
> We first clarify that our model outperforms previous work greatly because collision rate significantly improved. As demonstrated by the UniAD experiment in Table.4, DESIGN can significantly improve the collision rate by a relative margin of 25.8% (from 0.31 to 0.23), while achieving comparable L2 (from 1.03 to 1.05). L2 is not a good metric for safety-critical scenarios and a lower collision rate is preferred for safety over naive trajectory mimicking. So we argue this is a significant instead of slight improvement over the award-winning baseline UniAD.
>
> Q1: (1) **does not go beyond previous definitions in terms of object-level comprehensiveness**.
>
> We do use the pre-labelled 3D boxes from nuScenes, but to clarify, nuScenes annotate many boxes with the category of 'others' or 'pushable_pullable'. Our nuDESIGN dataset extends these annotations to detailed open-set entities like 'goose' or 'stroller'. So we argue that the object-level comprehensiveness is promoted.
>
> (2) **attach more** **meta** **data**.
>
> Directly attaching meta data, expressed in real numbers describing geometric entities like [-2.60m/s, 1.64m/s] or [-23.78m, 0.05m], to the caption would greatly challenge LLM since they are not natural language. We conduct experiments with this new version of captioning, where we attach meta data to the caption. The downstream planning modules follow the same protocol. This representation underperforms DESIGN as shown below. This experiment demonstrates that the performance gain brought by DESIGN is non-trivial and serves as another evidence of DESIGN's effectiveness.
>
> |                    | Input | C 1s     | C 2s     | C 3s     | Avg.     |
> | ------------------ | ----- | -------- | -------- | -------- | -------- |
> | Uniad              | 2D    | 0.05     | 0.17     | 0.71     | 0.31     |
> | DESIGN w/ metadata | 2D    | 0.07     | 0.15     | 0.61     | 0.28     |
> | DESIGN             | 2D    | **0.03** | **0.12** | **0.55** | **0.23** |
>
> Q2: (1) **most words are about objects' colors** .
>
> Indeed, we also include other important information like material, gloss, status (like 'clean') and most importantly, open-set entities. See Fig.8.
>
> (2) **drops the heading angle and accurate velocity**.
>
> To clarify, the DESIGN representation contains the moving status of 3D bounding boxes. For example, this is a ground truth caption in Fig.12: A stroller in the front left of ego car about 11 meters away is moving slowly to ego car in the crosswalk. As for whether accurate descriptions or rough natural language descriptions are better, the experiment above demonstrates that the 'metadata' representation underperforms DESIGN.
>
> Q3: **over-claim of 'scalable automatic labeling pipeline**.
>
> Indeed, our labeling pipeline is not fully automatic but an auto+manual one.  A fully automatic labeling pipeline would generate pseudo labels and is not what we want. We would like to clarify that our annotation method is easy to expand. With this pipeline, a worker only needs to decide what is right and what is wrong, instead of writing down every sentence. If the caption is considered right, it is included into the ground truth. If the caption is considered wrong, we just send the instance into the next round. Our annotation process takes three rounds, which are 0-20%, 20%-50%, and 50-100% of the data volume. We have counted the number of manual corrections in the three rounds, which was about 42.1%, 23.3% and 9.7% respectively.
>
> Q4: (1) **More baselines**.
>
> |                     | Input | C@0.25 | B-4@0.25 | M@0.25 | R@0.25 | C@0.5 | B-4@0.5 | M@0.5 | R@0.5 |
> | ------------------- | ----- | ------ | -------- | ------ | ------ | ----- | ------- | ----- | ----- |
> | Vote2Cap-DETR+MLE   | C+L   | 187.6  | 57.1     | 40.5   | 74.5   | 189.9 | 57.3    | 40.9  | 74.8  |
> | Vote2cap -DETR+SCST | C+L   | 198.3  | 58.0     | 41.9   | 75.6   | 200.1 | 58.4    | 42.1  | 76.1  |
> | Ours                | C+L   | 218.0  | 60.2     | 43.8   | 76.2   | 220.3 | 61.5    | 50.9  | 80.1  |
>
> We compare with the newest SOTA baseline Vote2Cap-DETR in the figure below. There are two settings: the first one exploits a standard cross entropy loss (denoted as maximum likelihood estimation, MLE) and the second one exploits a self-critical sequence training (denoted as SCST) that optimizes towards a better CIDEr metric. We can see that our model outperforms Vote2cap-DETR .
>
> (2) **2D dense captioning**.
>
> We show the results of 2D dense captioning on Tab.20.

---

> > ### Author Response · Authors · 2023-11-17
> > **Response to reviewer rzAd**
> >
> > Q5: **More ablation results**.
> >
> > (1) Training strategies：
> >
> > It is difficult to directly optimize the entire network, thus we utilize the widely used training strategy that divides the optimization process into several stages. We can see from the experiments below that the 3-stage training yields improvement significantly.
> >
> > |                                                          | C@0.25 | B-4@0.25 | M@0.25 | R@0.25 | C@0.5 | B-4@0.5 | M@0.5 | R@0.5 |
> > | -------------------------------------------------------- | ------ | -------- | ------ | ------ | ----- | ------- | ----- | ----- |
> > | One stage:Train all                                | 94.2   | 44.2     | 35.9   | 70.6   | 94.5  | 44.4    | 36.0  | 70.8  |
> > | Two stages:Train Detector+Train Decoder            | 152.4  | 50.9     | 38.2   | 72.9   | 153.8 | 51.0    | 38.4  | 73.1  |
> > | Three stages:Train Detector+Train Decoder+Train all | 158.7  | 51.2     | 38.4   | 73.1   | 160.2 | 51.4    | 38.5  | 73.3  |
> >
> > (2) Different language decoders:
> >
> > We conduct experiments with different language decoders and we observe that LLAMA achieves the best performance.
> >
> > |        | C@0.25 | B-4@0.25 | M@0.25 | R@0.25 | C@0.5  | B-4@0.5 | M@0.5 | R@0.5 |
> > | ------ | ------ | -------- | ------ | ------ | ------ | ------- | ----- | ----- |
> > | S&T[1] | 116.3  | 45.2     | 38.0   | 70.9   | 116.5  | 45.4    | 38.3  | 71.0  |
> > | GPT2   | 137.0  | 47.7     | 37.8   | 71.1   | 5137.2 | 47.8    | 38.1  | 71.2  |
> > | LLAMA  | 158.7  | 51.2     | 38.4   | 73.1   | 160.2  | 51.4    | 38.5  | 73.3  |
> >
> > [1] Show and Tell: A Neural Image Caption Generator
> >
> > Q6 Q7 Q8: **Some mistakes.** We thank the reviewer for this suggestion. 6. We have removed the end-to-end AD part in related work. 7. We have fixed this mistake. 8. We have included these three formulas into the paragraph.
> >
> > Q9: Yes. The size of object proposals boxes is N\*9 and the size of features is N\*C.
> >
> > Q10: **how to tell**. We should clarify this scenario. Indeed, we can see from the back of the camera that there is a bus behind ego car. The LLM infers that he is about to stop at the bus stop so the ego car should keep driving to avoid the bus stop. However, the uniad slows down and has already driven into the bus stop, resulting in a collision with the bus.

---

> > > ### Comment · Reviewer_rzAd · 2023-11-19
> > > **Response to authors' rebuttal**
> > >
> > > I really appreciate the authors' detailed rebuttal. I have also looked into other reviewers' comments and authors' replies. I acknowledge that using LLM for autonomous driving is a hot topic and the annotation takes a huge amount of effort. However, I still have a few concerns and comments.
> > >
> > > - After more reading, I realized that the paper at the current stage lacks a lot of details, including but not limited to the following,
> > >   - Whether the captioning baseline methods are finetuned or directly inferred in a zero-shot manner.
> > >   - The UniAD experiment details such as how many epochs for the training and if training/finetuning the full model. I also see that the velocity is attached in the language prompt in Fig. 14 but UniAD doesn't. This could potentially lead to unfair comparisons.
> > >   - In order to show the reasoning ability with language, how about showing the generated language together with downstream results?
> > > - In reviewer a5o2's Q1, the improvement on BEVFormer-base is very minor. The proposed method has finetuned BEVFormer with the language decoder in the final training stage. Training BEVFormer longer could probably lead to much better performance. Besides, the ablation in my Q5 further indicates that the proposed task and method highly depend on the detector. The question in my previous Q1 is that I feel like the contribution of the task is to attach more metadata (eg, color) with detected objects.
> > > - I found a lot of citation format errors: space before citations, and wrong use of citep/citet (eg, Sec 5.4 proposed in Li et al. / Sec 6.5 CLIP embedding)

---

> > > > ### Author Response · Authors · 2023-11-22
> > > > **Response to reviewer rzAd**
> > > >
> > > > We would like to thank rzAd for the response. Here we respond to the concerns one by one:
> > > >
> > > > Q11.1: **Whether the captioning baseline methods are finetuned or directly inferred in a zero-shot manner**.
> > > >
> > > > To clarify, the captioning baselines in Table.2 and Table.3 are fine-tuned on the training set of nuDESIGN, instead of directly inferred with models trained on ScanNet. In Table.2, it is shown that, when only Lidar is used as input, Scan2Cap's performance is poor despite it is fine-tuned on nuDESIGN.
> > > >
> > > > Q11.2.1: **How many epochs for the training and if training/finetuning the full model** .
> > > >
> > > > There are two stages in original UniAD. The first stage trains the perception and prediction modules. The second end-to-end stage trains the full model for end-to-end planning. In our experiment, we only train the second end-to-end planning stage (with config "stage2_e2e") while loading weights trained in the first stage. In our experiment, modules pre-trained in the first stage are also end-to-end fine-tuned. In the two experiments in Table.4, we train the models for 20 epochs, which is the same as the original second stage setting in UniAD.
> > > >
> > > > Q11.2.2: **Unfair comparisons**.
> > > >
> > > > In UniAD, the past trajectory of ego-car is used as input to the motion prediction module. In our experiment, the speed information (e.g., 'The ego car's ego_car_velocity is 3.32 m/s') is also derived from the past trajectory of ego-car. So we argue that the comparison is fair because we exploit the same information source.
> > > >
> > > > Q11.3: **Showing the generated language together with downstream results**.
> > > >
> > > > We thank the reviewer for the suggestion. We have displayed the generated language of LLM and downstream planning results together in Fig.19.
> > > >
> > > > Q12.1: **The improvement on BEVFormer-base is very minor. Training BEVFormer longer could probably lead to much better performance.**
> > > >
> > > > We conduct experiments with BEVFormer for 48 epochs (compared with previous 24 epochs). Training BEVFormer longer still under-performs DESIGN-Former for BEVFormer-tiny and BEVFormer-small, as shown in in the following Tables. For BEVFormer-base, training longer leads to slightly better results than ours.
> > > >
> > > > | (early stop epoch / total epoch) | mAP      | NDS      |
> > > > | -------------------------------- | -------- | -------- |
> > > > | BEVFormer-tiny (24/48 epochs)    | 22.3     | 32.4     |
> > > > | BEVFormer-tiny (36/48 epochs)    | 25.2     | 36.8     |
> > > > | BEVFormer-tiny (48/48 epochs)    | 27.0     | 38.5     |
> > > > | Ours-tiny                        | **27.6** | **39.4** |
> > > >
> > > >
> > > >
> > > > | (early stop epoch / total epoch) | mAP      | NDS      |
> > > > | -------------------------------- | -------- | -------- |
> > > > | BEVFormer-small (24/48 epochs)   | 31.1     | 42.0     |
> > > > | BEVFormer-small (36/48 epochs)   | 34.7     | 45.8     |
> > > > | BEVFormer-small (48/48 epochs)   | 36.8     | 47.8     |
> > > > | Ours-small                       | **37.6** | **48.7** |
> > > >
> > > >
> > > >
> > > > | (early stop epoch / total epoch) | mAP      | NDS      |
> > > > | -------------------------------- | -------- | -------- |
> > > > | BEVFormer-base(24/48 epochs)     | 34.0     | 44.2     |
> > > > | BEVFormer-base(36/48 epochs)     | 38.5     | 48.2     |
> > > > | BEVFormer-base(48/48 epochs)     | **42.1** | **51.9** |
> > > > | Ours-base                        | 41.8     | 51.8     |
> > > >
> > > > Q12.2: **The contribution of the task is to attach more metadata (eg, color) with detected objects.**
> > > >
> > > > We still believe that our contribution is summarizing important target status in natural language (as a novel middleware), instead of only attaching more metadata. Specifically, we show an experiment in Q1 response that 'DESIGN w/ metadata' underperforms DESIGN in the UniAD experiment.
> > > >
> > > > Q13: **Citation format errors**.
> > > >
> > > > We thank the reviewer for pointing out the mistakes in the citation. We have fixed the usage of citep/citet in the new version.

---

### Official Review · Reviewer_a5o2 · 2023-10-30

**Soundness:** 2 fair
**Presentation:** 2 fair
**Contribution:** 2 fair
**Rating:** 3
**Confidence:** 5

**Summary:**

The authors of this study investigate intermediate representations that connect perception and planning. They propose that a good representation should be comprehensive and standardized. However, existing representations are not compatible with recent language foundation models that have demonstrated exceptional reasoning capabilities. In order to address this, the authors explore an intermediate representation called DESIGN (3D dense captioning beyond nouns) to determine if it can enhance the intelligence and safety of autonomous vehicles. They develop a scalable automatic labeling pipeline to create a large dataset called nuDESIGN, which consists of 2,300k descriptions for 1,165k objects in 850 scenes. Additionally, the authors contribute DESIGN-former, a query-based network architecture that fine-tunes adapters on LLaMa. DESIGN-former outperforms existing dense captioning baselines by a significant margin. Lastly, the authors present a pilot study that demonstrates the impact of this new middleware representation on an end-to-end driving model.

**Strengths:**

1. Importance of the work: The authors highlight an important direction for learning representations that connect perception and planning. While the community explores different intermediate representations such as bounding boxes, semantic maps, and occupancy grids, a consensus on the form of representations has not been reached. This topic will be of interest to the community studying representation learning and learning-based driving models.
2. Challenge of the work: In this work, the authors aim to define a comprehensive and standard representation that also possesses reasoning ability. The objective is challenging and requires a comprehensive evaluation to demonstrate the effectiveness of the proposed representation.
3. A new intermediate representation called DESIGN (3D dense captioning beyond nouns): The representation consists of five different components: Appearance, Direction, Distance, Motion, and Road Map. The authors propose a specific form for the representation that covers a wide range of information. Furthermore, these pieces of information (direction, distance, motion, and road map) can be extracted from existing representations. The reviewer finds DESIGN to be a standard representation that defines a clear format for forming the representation.
4. A new dataset called nuDESIGN has been collected using the proposed automatic pipeline. nuDESIGN contains 2300k descriptions for 1165k objects in 850 scenes. To the best of the reviewer's knowledge, this is the largest dataset with language captions. Additionally, the dataset enables 2D/3D dense captioning in traffic scenes. As far as the reviewer know, the size of the dataset is the largest in traffic scenes.
5. Method and Experiment: On the proposed dataset nuDESIGN, the proposed method DESIGN-former achieves favorable results in 2D/3D dense captioning, compared to

**Weaknesses:**

1. My main concern is that the paper does not clearly explain the effectiveness of DESIGN for downstream tasks. The reviewer believes that this should be the main focus of the paper, as mentioned in the last sentence of the first paragraph of the introduction. However, most of the evaluations focus on 2D/3D dense captioning, rather than the original motivation of enabling smarter and safer autonomous vehicles. I recommend that the authors focus on demonstrating that the proposed nuDESIGN is a "new addition" for 2D/3D dense captioning in traffic scenes.
2. The experiment on the end-to-end driving model is not convincing. While the authors show an approach to link the language model with the end-to-end model, the results do not explicitly demonstrate that the learned representations enable human-like reasoning. I suggest that the authors consider scenarios studied in ReasonNet (Shao et al., "ReasonNet: End-to-End Driving with Temporal and Global Reasoning, CVPR 2023"), which require explicit modeling of socially semantic spatial-temporal relationships.
3. The smoother trajectory shown in Figure 15 is also not convincing, considering the significant effort in creating a dataset and large language models. The results could be generated through post-processing of the generated trajectories. Is it necessary to include the proposed method? I suggest that the authors consider scenarios that are challenging for existing methods.
4. Limitation of DESIGN. While the middleware captures a wide range of information, two critical aspects are missing: uncertain intents of an object and the ability to facilitate forecasting. These aspects have been extensively studied and found to be valuable in the field of autonomous driving. I would like to learn the authors' thoughts on these aspects.
5. There is a lack of comparisons with existing baselines for 3D dense captioning. For example:
    1. Chen et al., "End-to-End 3D Dense Captioning with Vote2Cap-DETR, CVPR 2023."
    2. Cai et al., "3djcg: A unified framework for joint dense captioning and visual grounding on 3d point clouds, CVPR 2022."
    3. Chen et al., "D3net: A speaker-listener architecture for semi-supervised dense captioning and visual grounding in rgb-d scans, ECCV 2022."

    Without a direct comparison, we cannot conclude whether the proposed method achieves state-of-the-art performance.

**Questions:**

The reviewer identify five major concerns in the Weakness section and would like to know the authors' thoughts on these points. Please answer each concern in the rebuttal stage. The reviewer will respond according to the authors' rebuttal in the discussion phase.

---

> ### Author Response · Authors · 2023-11-17
> **Response to reviewer a5o2**
>
> We thank a5o2 for these constructive feedbacks. Here we respond to five concerns one by one:
>
> Q1: **Effectiveness for downstream tasks**.
>
> We would like to remark on the effectiveness of the DESIGN middleware from three perspectives. **Firstly**, as demonstrated by the end-to-end driving experiment in Table.4, DESIGN can significantly decrease the collision rate by a relative margin of 25.8% (from 0.31 to 0.23), while achieving comparable L2 (from 1.03 to 1.05). We argue that collision rate is a more meaningful indicator than L2 for autonomous driving, so this result is significant enough to demonstrate the effectiveness of DESIGN. **Secondly**, we hypothesize (with all due respect) that a5o2 may feel the UniAD experiment does not contain enough datapoints. In this regard, we provide another datapoint in Table.4 that directly turns the exact real numbers into natural language descriptions. This representation underperforms DESIGN. This datapoint demonstrates that the performance gain brought by DESIGN is non-trivial and serves as another evidence of DESIGN's effectiveness. **Thirdly**, we evaluate in another setting that shows DESIGN can also improve detection performance. The results below show that the DESIGN-former outperforms BEVFormer[1] in mAP and NDS, suggesting that the cues from language helps to regularize the middle representation of BEV features and improve the performance of the bounding box detection. This serves as another evidence that DESIGN has some superiority over an existing middleware that is a subset of DESIGN (i.e., 3D bounding boxes).
>
> |                     | mAP      | NDS      |
> | ------------------- | -------- | -------- |
> | BEVFormer-tiny      | 25.2     | 35.4     |
> | DESIGN-former-tiny  | **27.6** | **39.4** |
> | BEVFormer-small     | 37.0     | 47.9     |
> | DESIGN-former-small | **37.6** | **48.7** |
> | BEVFormer-base      | 41.6     | 51.7     |
> | DESIGN-former-base  | **41.8** | **51.8** |
>
> Q2: **Do not explicitly demonstrate the learned representations enable human-like reasoning**.
>
> We would like to respond to this concern from three perspectives: (1) The ReasonNet paper has been cited and discussed in an updated version, but annotating the ReasonNet dataset with DESIGN is not feasible in the rebuttal window. (2) Our UniAD experiment (Table.4) has a module that requires interaction with GPT-3.5, which is demonstrated in Fig.14 and Fig.16. We hypothesize (with all due respect) that a5o2 feels the cases shown in these two figures are not human-like enough, so we provide two more cases into Fig.18.
>
> Q3: **More challenging scenarios**.
>
> We thank the reviewer for this suggestion. We provide more visualization results of our driving model in challenging scenarios in Fig.19. For example, in the last example of Fig.19, a white truck in ego lane blocks the way. The UniAD tries to merge to the right lane and collides with the silver car on the right lane, while our driving model stops to avoid the collision with right car.
>
> Q4: **Uncertain intents of an object and the ability to facilitate forecasting**.
>
> To clarify, the DESIGN representation contains the moving status of 3D bounding boxes. For example, this is a ground truth caption in Fig.12: A stroller in the front left of ego car about 11 meters away is moving slowly to ego car in the crosswalk.
>
> Here 'moving slowly to ego car' covers the moving trend of the object. As for 'uncertain intents', we think the rough descriptions of 'slowly' or 'quickly' have intrinsically modeled the uncertainty.
>
> Q5: **Comparisons with more baselines**.
>
> |                      | Input | C@0.25 | B-4@0.25 | M@0.25 | R@0.25 | C@0.5 | B-4@0.5 | M@0.5 | R@0.5 |
> | -------------------- | ----- | ------ | -------- | ------ | ------ | ----- | ------- | ----- | ----- |
> | Vote2cap + MLE       | C+L   | 187.6  | 57.1     | 40.5   | 74.5   | 189.9 | 57.3    | 40.9  | 74.8  |
> | Vote2cap with + SCST | C+L   | 198.3  | 58.0     | 41.9   | 75.6   | 200.1 | 58.4    | 42.1  | 76.1  |
> | Ours                 | C+L   | 218.0  | 60.2     | 43.8   | 76.2   | 220.3 | 61.5    | 50.9  | 80.1  |
>
> We thank a5o2 for pointing out these references and we have cited them in the updated paper. We compare with the newest SOTA baseline Vote2Cap-DETR in the figure below. There are two settings: the first one exploits a standard cross entropy loss (denoted as maximum likelihood estimation, MLE) and the second one exploits a self-critical sequence training (denoted as SCST) that optimizes towards a better CIDEr metric. As for 3DJCG and D3Net, since they are older baselines and involve a substantial modification to the dataloader for joint captioning and grounding, we have not yet evaluated in the rebuttal window.

---

> > ### Comment · Reviewer_a5o2 · 2023-11-23
> > **Appreciate the authors' detailed rebuttal**
> >
> > I appreciate the authors' detailed rebuttal. The following are my responses to the rebuttal.
> >
> > 1. I agree with the authors that collision is a better metric for evaluation. However, I am not clear if the improvement is due to the proposed middleware. Specifically, the authors do not show that the proposed middleware can improve those failure cases in UniAD. If the authors can show the evidence, it will be more convincing. Regarding the additional experiments on detection, I appreciate the authors' efforts. However, I am not convinced that the proposed middleware is a better choice than other middleware on detection. Without comparisons, we cannot conclude its effectiveness.
> >
> > 2. Thanks for the additional results.
> >
> > 3. Thanks for sharing additional results! I am expecting more challenging scenarios, similar to ReasonNet, where AD agents are involved in interactive or occluded scenarios. It is more convincing to show the value of "reasoning."
> >
> > 4. To clarify, I am curious if the proposed model can infer situations where interactive agents have multiple intents, e.g., "slowly moving" or "aggressively turning right." Do the authors find errors or collisions when the model makes incorrect intent reasoning?
> >
> > 5.  Thanks for sharing additional results. I wonder if the authors finetune Vote2cap on nuScenes?

---

### Author Response · Authors · 2023-11-19
**Asking for feedbacks.**

Dear meta-reviewers and reviewers:

We express our gratitude to the meta-reviewers and reviewers for their valuable time dedicated to our manuscript. In response to these insightful feedback, we have diligently conducted new experiments and provided additional clarifications. We kindly request their consideration in reviewing these revisions at their earliest convenience. Your feedback on any remaining concerns would be immensely helpful, enabling us to offer further enhancements before the deadline on Nov. 22nd.

Sincerely,
Authors.

---

### Meta-Review · Area_Chair_VHC4 · 2023-12-09

**Metareview:**

This paper's primary contribution is a dense captioning dataset for autonomous driving, labeled with 2D or 3D bounding boxes and their captions, a task on which the paper does better than related works in this area. The paper claims that it is useful for downstream planning tasks for autonomous driving, by showing for example that collision rates (for 1-3 seconds in the future) on nuScenes is reduced compared to UniAD. Bounding box detection and tracking is clearly a fundamental problem in autonomous driving, but unfortunately, the paper does not provide a convincing argument for exactly how the captions could be useful for self-driving behavior.

**Justification For Why Not Higher Score:**

I think the paper needs one more iteration to make an argument for how dense captions could be used by a driving policy that relies on a scene representation.

**Justification For Why Not Lower Score:**

N/A

---

### Decision · Program_Chairs · 2024-01-16

Reject